# ON USING ADMISSIBLE BOUNDS FOR LEARNING FORWARD SEARCH HEURISTICS

## ABSTRACT

In recent years, there has been growing interest in utilizing modern machine learning techniques to learn heuristic functions for forward search algorithms. Despite this, there has been little theoretical understanding of *what* they should learn, *how* to train them, and *why* we do so. This lack of understanding has resulted in the adoption of diverse training targets (suboptimal vs optimal costs vs admissible heuristics) and loss functions (e.g., square vs absolute errors) in the literature. In this work, we focus on how to effectively utilize the information provided by admissible heuristics in heuristic learning. We argue that learning from poly-time admissible heuristics by minimizing mean square errors (MSE) is not the correct approach, since its result is merely a noisy, inadmissible copy of an efficiently computable heuristic. Instead, we propose to model the learned heuristic as a *truncated* gaussian, where admissible heuristics are used not as training targets but as lower bounds of this distribution. This results in a different loss function from the MSE commonly employed in the literature, which implicitly models the learned heuristic as a gaussian distribution. We conduct experiments where both MSE and our novel loss function are applied to learning a heuristic from optimal plan costs. Results show that our proposed method converges faster during training and yields better heuristics, with 40% lower MSE on average.

## 1 INTRODUCTION

Motivated by the success of Machine Learning (ML) approaches in various decision making tasks (Mnih et al., 2015; Silver et al., 2016), an increasing number of papers are tackling the problem of learning a heuristic function for forward state space search in recent years. Despite this interest, there has been little theoretical understanding of *what* these systems should learn, *how* to train them and *why* we do so. As a result, heuristic learning literature has adopted many different training targets (corresponding to either admissible heuristics (Shen et al., 2020), suboptimal solution costs (Arfaee et al., 2011; Ferber et al., 2022; Marom & Rosman, 2020) or optimal solution costs (Ernandes & Gori, 2004; Shen et al., 2020)) and training losses (e.g., square errors (Shen et al., 2020), absolute errors (Ernandes & Gori, 2004) and piecewise absolute errors (Takahashi et al., 2019)).

In this work, we try to answer these questions from a statistical lens, focusing on how to effectively utilize admissible heuristics in the context of heuristic learning. We argue that learning from poly-time admissible heuristics, such as $h^{\text{LMcut}}$ (Helmert & Domshlak, 2009), by minimizing mean square errors (MSE) does not provide any practical benefits, since its result is merely a noisy, inadmissible copy of a heuristic that is already efficient to compute. Then, if admissible heuristics should not be used as training targets, how can we leverage them? In order to answer this question, we first analyze the statistical implications behind the commonly used loss function, the MSE, which implicitly models the learned heuristic as a Gaussian distribution. Nonetheless, we contend that a better modeling choice for heuristics is given by the *Truncated* Gaussian distribution (Fig. 1), due to the existence of *bounds* on the values a heuristic can take (e.g., heuristics never take on negative values).

The main contribution of this paper is a theoretically-motivated, statistical method for exploiting the information encoded by admissible heuristics in the heuristic learning setting. We propose to model the learned heuristic as a Truncated Gaussian, where an admissible heuristic provides the lower bound of this distribution. This modeling choice results in a loss function to be minimized that is different from the standard MSE loss. We conduct extensive experimentation where both loss functions are

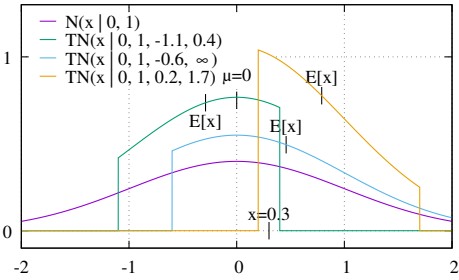

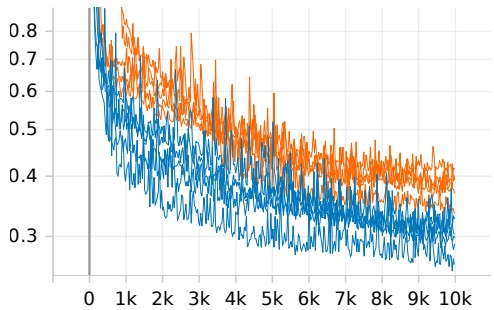

Figure 1: The PDFs of Truncated Gaussian distributions $p(\mathrm{x}) = \mathcal{TN}(\mu = 0, \sigma = 1, l, u)$ with several lower/upper bounds $(l, u)$. In the heuristic learning setting, x is the optimal solution cost $h^*$ sampled from the dataset and $\mu = \mu_\theta(s)$ is the prediction associated with a state $s$. The $(l, u) = (0.2, 1.7)$ variant (yellow) shows that the mean $\mathbb{E}_{p(\mathrm{x})}[\mathrm{x}]$, which we use as the search heuristic, respects the bounds $(l, u)$ even when the predicted $\mu = 0$ lies outside $(l, u)$.

Figure 2: Comparison of the training curve ($x$-axis: training step) for the validation MSE loss ($y$-axis, logarithmic) between Gaussian (orange) and Truncated Gaussian (blue) models on logistic domain, independent runs recorded on 5 random seeds each. The losses converge faster for the latter due to the additional information provided by the admissible lower bound $l = h^{\mathrm{LMcut}}$.

applied to learning heuristics from optimal plan costs in several classical planning domains. Results show that those methods which model the learned heuristic as a Truncated Gaussian learn faster and result in more accurate heuristics, with 40% lower MSE on average, than those which model it as an ordinary Gaussian, i.e., minimize an MSE loss during training.

## 2 BACKGROUND: CLASSICAL PLANNING AND HEURISTICS

We define a propositional STRIPS Planning problem as a 4-tuple $\langle P, A, I, G \rangle$ where $P$ is a set of propositional variables, $A$ is a set of actions, $I \subseteq P$ is the initial state, and $G \subseteq P$ is a goal condition. Each action $a \in A$ is a 4-tuple $\langle \mathrm{PRE}(a), \mathrm{ADD}(a), \mathrm{DEL}(a), \mathrm{COST}(a) \rangle$ where $\mathrm{COST}(a) \in \mathbb{Z}^{0+}$ is a cost, $\mathrm{PRE}(a) \subseteq P$ is a precondition and $\mathrm{ADD}(a), \mathrm{DEL}(a) \subseteq P$ are the add-effects and delete-effects. A state $s \subseteq P$ is a set of true propositions (all of $P \setminus s$ are false), an action $a$ is *applicable* when $s \supseteq \mathrm{PRE}(a)$ (read: $s$ *satisfies* $\mathrm{PRE}(a)$), and applying action $a$ to $s$ yields a new successor state $a(s) = (s \setminus \mathrm{DEL}(a)) \cup \mathrm{ADD}(a)$.

The task of classical planning is to find a sequence of actions called a *plan* $(a_1, \cdots, a_n)$ where, for $1 \leq t \leq n$, $s_0 = I$, $s_t \supseteq \mathrm{PRE}(a_{t+1})$, $s_{t+1} = a_{t+1}(s_t)$, and $s_n \supseteq G$. A plan is *optimal* if there is no plan with lower *cost-to-go* $\sum_t \mathrm{COST}(a_t)$. A plan is otherwise called *satisficing*. In this paper, we assume unit-cost: $\forall a \in A; \mathrm{COST}(a) = 1$.

A domain-independent heuristic function $h$ in classical planning is a function of a state $s$ and the problem $\langle P, A, I, G \rangle$. It returns an estimate of the shortest (optimal) path cost from $s$ to one of the goal states (states that satisfy $G$), typically through a symbolic, non-statistical means including problem relaxation and abstraction. Notable state-of-the-art functions include $h^{\mathrm{LMcut}}, h^{\mathrm{FF}}, h^{\max}, h^{\mathrm{add}}, h^{\mathrm{GC}}$ Helmert & Domshlak (2009); Hoffmann & Nebel (2001); Bonet & Geffner (2001); Fikes et al. (1972). The optimal cost to go, or a *perfect heuristic*, is denoted by $h^*$. *Admissible* heuristics are those that never overestimate it, i.e., $\forall s; 0 \leq h(s) \leq h^*(s)$.

## 3 TASK: SUPERVISED LEARNING FOR HEURISTICS

This section explains the basic statistical background behind the commonly used MSE loss, including the assumptions that are implicitly made when it is used for training a model.

Let $p^*(\mathrm{x})$ be the unknown ground-truth probability distribution of an observable random variable(s) x. Given a dataset $\mathcal{X} = \left\{ x^{(1)}, \ldots, x^{(N)} \right\}$ of $N$ data points, we denote an empirical data distribution as $q(\mathrm{x})$, which draws samples from $\mathcal{X}$ uniformly and can be regarded as a uniform mixture of dirac's delta distributions (Eq. 1), which is entirely different from $p^*(\mathrm{x})$. Our goal is to obtain an estimate $p(\mathrm{x})$ that resembles $p^*(\mathrm{x})$ as closely as possible. Under the *Maximum Likelihood Estimation* (MLE) framework, it is assumed that $p^*(\mathrm{x})$ is equal to the $p(\mathrm{x})$ that maximizes its expectation over $q(\mathrm{x})$. In other words, MLE tries to maximize the probability $p(x)$ of observing each data point $x \sim q(\mathrm{x})$:

$$q(\mathrm{x}) = \sum_{\mathrm{i}} q(\mathrm{x}|\mathrm{i})q(\mathrm{i}) = \sum_{i=1}^{N} \delta(\mathrm{x} = x^{(i)}) \cdot \frac{1}{N} \tag{1}$$

$$p^*(\mathrm{x}) = \arg\max_p \mathbb{E}_{q(\mathrm{x})} p(\mathrm{x}) = \arg\max_p \mathbb{E}_{q(\mathrm{x})} \log p(\mathrm{x}) = \arg\min_p \mathbb{E}_{q(\mathrm{x})} - \log p(\mathrm{x}) \tag{2}$$

where $\mathbb{E}_{q(\mathrm{x})}$ is estimated by Monte-Carlo in practice.

Typically, we assume $p^*(\mathrm{x})$ and $p(\mathrm{x})$ are of the same family of functions parameterized by $\theta$, such as a set of neural network weights, i.e., $p^*(\mathrm{x}) = p_{\theta^*}(\mathrm{x})$, $p(\mathrm{x}) = p_\theta(\mathrm{x})$. This makes MLE a problem of finding the $\theta$ maximizing $\mathbb{E}_{q(\mathrm{x})} p_\theta(\mathrm{x})$. In practice, we typically minimize a *loss* such as the *negative log likelihood* (NLL) $-\log p(\mathrm{x})$, since $\log$ is monotonic and preserves the optima $\theta^*$ (Eq. 2). We further assume $p(\mathrm{x})$ to follow a specific distribution such as a Gaussian distribution:

$$p(\mathrm{x}) = \mathcal{N}(\mu, \sigma) = \frac{1}{\sqrt{2\pi\sigma^2}} e^{-\frac{(\mathrm{x}-\mu)^2}{2\sigma^2}}. \tag{3}$$

We emphasize that *the choice of the distribution determines the loss*. When the model designer assumes $p(\mathrm{x}) = \mathcal{N}(\mu, \sigma)$, then the NLL is a shifted and scaled squared error:

$$-\log p(\mathrm{x}) = \frac{(\mathrm{x}-\mu)^2}{2\sigma^2} + \log\sqrt{2\pi\sigma^2}. \tag{4}$$

Similarly, a Laplace distribution $L(\mu, b) = \frac{1}{2b} e^{-\frac{|\mathrm{x}-\mu|}{b}}$ results in a shifted and scaled absolute error $\frac{|\mathrm{x}-\mu|}{b} + \log 2b$ as the NLL.

It can be seen that the NLL derived from a Gaussian distribution (Eq. 4) is similar but not exactly equivalent to the MSE. The reason is that most methods, when provided with an input, do not return a probability distribution $p(\mathrm{x})$ but rather a single prediction (referred to as a *point estimate*) which represents the entire $p(\mathrm{x})$. A point estimate can be any statistic of central tendency, such as the mean, median or mode. In the case of a Gaussian $p(\mathrm{x}) = N(\mu, \sigma)$, these three values are identical to $\mu$. Therefore, since $\sigma$ does not affect the point estimate of $N(\mu, \sigma)$, the NLL loss can be simplified by setting a fixed $\sigma = \frac{1}{\sqrt{2}}$ and ignoring the second term of Eq. 4 (which becomes a constant), resulting in a square error $(\mathrm{x} - \mu)^2$. The expectation $\mathbb{E}_{q(\mathrm{x})}(\mathrm{x}-\mu)^2$ of this error can be estimated by Monte-Carlo, by sampling $N$ data points $x \sim q(\mathrm{x})$ and averaging the loss $(x-\mu)^2$ of each sample $x$ resulting in the standard *mean* square error MSE: $\frac{1}{N}\sum(x-\mu)^2$. In contrast, *distributional estimation* returns $p(\mathrm{x})$ without simplifications and thus $p(\mathrm{x}) = N(\mu, \sigma)$ must predict the values for both $\mu$ and $\sigma$.

The MLE framework can be applied to the supervised heuristic learning setting as follows. Let $q(\mathrm{s}, \mathrm{x})$ be the empirical data distribution, where s is a random variable representing a state-goal pair (from now on, we will implicitly assume that states s also contain goal information) and x a random variable representing the cost-to-go (regardless of whether it corresponds to a heuristic estimate, optimal or suboptimal cost). Then, the goal is to learn $p^*(\mathrm{x} \mid \mathrm{s})$ where:

$$p^*(\mathrm{x} \mid \mathrm{s}) = \arg\max_p \mathbb{E}_{q(\mathrm{s},\mathrm{x})} p(\mathrm{x}|\mathrm{s}) \tag{5}$$

$$p(\mathrm{x} \mid \mathrm{s}) = \mathcal{N}(\mathrm{x} \mid \mu = \mu_\theta(\mathrm{s}), \sigma = \tfrac{1}{\sqrt{2}}), \tag{6}$$

and $\mu_\theta(\mathrm{s})$ is the main body of the learned model, such as a neural network parameterized by the weights $\theta$. Eq. 6 holds because the network predicting the mean $\mu = \mu(\mathrm{s})$ is deterministic, i.e., a direc's delta $p(\mu \mid \mathrm{s}) = \delta(\mu = \mu(\mathrm{s}))$ that assigns a probability of 0 to every $\mu \neq \mu(\mathrm{s})$. Supervised heuristic learning with distributional estimates is formalized similarly. The only difference is that an additional model (e.g. a neural network) with parameters $\theta_2$ predicts $\sigma$:

$$p(\mathrm{x} \mid \mathrm{s}) = \mathcal{N}(\mathrm{x} \mid \mu = \mu_{\theta_1}(\mathrm{s}), \sigma = \sigma_{\theta_2}(\mathrm{s})). \tag{7}$$

## 3.1 THE PRINCIPLE OF MAXIMUM ENTROPY

In the discussion above, we decided to model $p(\mathrm{x})$ as a Gaussian distribution. Nonetheless, we may wonder if this is the correct choice. The *principle of maximum entropy* (Jaynes, 1957) states that $p(\mathrm{x})$ should be modeled as the maximum entropy (*max-ent*) distribution among all those that satisfy our constraints or assumptions, where entropy is defined as $\mathbb{E}_{p(x)}\langle -\log p(x)\rangle$. A set of constraints defines its corresponding max-ent distribution which, being the *most random* among those that satisfy those constraints, does not encode any more assumptions than those associated with the given constraints. Conversely, a non max-ent distribution implicitly encodes additional or different assumptions and, thus, can result in an accidental, potentially harmful bias. For example, if we believe that our random variable x has a finite mean, a finite variance and a support/domain equal to $\mathbb{R}$, then it should be modeled as a Gaussian distribution, since it is the max-ent distribution among all those that satisfy these three constraints.

Therefore, in order to select an appropriate distribution for $p(\mathrm{x})$, the model designer should first devise a reasonable set of constraints and then model $p(\mathrm{x})$ as the max-ent distribution that satisfies those constraints which, in turn, will determine the particular NLL loss function to minimize for training the model. This paper tries to follow this principle as faithfully as possible.

## 4 UTILIZING BOUNDS FOR LEARNING

In the previous section, we provided some statistical background on heuristic learning. We now leverage this background to analyze many of the decisions taken in the existing literature, sometimes unknowingly. Among the different aspects of heuristic learning, we put particular focus on the best way of utilizing the information provided by admissible heuristics during training.

We previously explained that the heuristic to be learned is modeled as a probability distribution (e.g., a Gaussian), instead of a single value. The reason behind this is that the ML model is unsure about the true heuristic value $h^*$ associated with a state $s$. When it predicts $\mu$, it believes not only that $\mu$ is the most likely value for $h^*$ but also that other values are still possible. The uncertainty of this prediction is given by $\sigma$: The larger this parameter is, the more unsure the model is about its prediction. The commonly used MSE loss is derived from the ad-hoc assumption that $\sigma$ is fixed, i.e., it does not depend on $s$. This would mean that the model is equally certain (or uncertain) about $h^*$ for every state $s$, which is not a realistic assumption in most scenarios. For example, it is generally more difficult to accurately predict $h^*$ for those states that are further from the goal due to the unknown obstacles between the current state and the goal states. Therefore, the model should predict $\sigma$ in addition to $\mu$, i.e., it should output a distributional estimate of $h^*$ instead of a point estimate, which should improve the speed of convergence.

Another crucial decision involves selecting *what* to learn, i.e., the target / ground truth to use for the training. It is easy to see that training a model on a dataset containing a practical (i.e., computable in polynomial time) admissible heuristic such as $h^{\mathrm{LMcut}}$ does not provide any practical benefits. Even in the best case, we will simply obtain a noisy, lossy, inadmissible copy of a heuristic that is already efficient to compute. Therefore, in order to learn a heuristic that outperforms these poly-time admissible heuristics, i.e., achieve a *super-symbolic benefit* from learning, it is imperative to train the model on data of better quality. For instance, it can be trained on the $h^+$ heuristic, as proposed in Shen et al. (2020), or even on optimal solution costs $h^*$, although obtaining these datasets may prove computationally expensive in practice. Nonetheless, by training on these targets, we can aspire to learn a heuristic that outperforms the poly-time admissible heuristics, although at the cost of the loss of admissibility.

If poly-time admissible heuristics are not useful as training targets, are they completely useless in learning a heuristic? Intuitively this should not be the case, given the huge success of heuristic search where they provide a strong search guidance toward the goal. Our main quiestion is then about *how* we should exploit the information they provide. To figure it out, we must revise the assumption we made (by using squared errors) about $\mathrm{x} = h^*$ following a Gaussian distribution $\mathcal{N}(\mu, \sigma)$. The issue with this assumption is that a Gaussian distribution assigns a non-zero probability $p(x)$ to every $x \in \mathbb{R}$, but we actually know that $h^*$ cannot take some values. In particular, given some admissible heuristic like $h^{\mathrm{LMcut}}$, we know that the inequality $h^{\mathrm{LMcut}} \leq h^*$ holds for every state; therefore

$p(x) = 0$ when $x < h^{\text{LMcut}}$. Analogously, if for some state $s$ we know the cost $h^{sat}$ of a satisficing (non-optimal) plan from $s$ to the goal, then $h^{sat}$ acts as an *upper bound* of $h^*$.

According to the principle of maximum entropy, which serves our *why*, if we have a lower $l$ and upper $u$ bound for $h^*$, then we should model $h^*$ using the max-ent distribution with finite mean, finite variance, and a support equal to $(l, u)$. The max-ent distribution that satisfies all these constraints is the *Truncated Gaussian* distribution $\mathcal{TN}(\mathrm{x}|\mu, \sigma, l, u)$ (Dowson & Wragg, 1973), formalized as Eq. 8:

$$\mathcal{TN}(\mathrm{x}|\mu, \sigma, l, u) = \begin{cases} \frac{1}{\sigma} \frac{\phi(\frac{\mathrm{x}-\mu}{\sigma})}{\Phi(\frac{u-\mu}{\sigma}) - \Phi(\frac{l-\mu}{\sigma})} & l \le \mathrm{x} \le u \\ 0 & \text{otherwise.} \end{cases}$$

$$\phi(\mathrm{x}) = \frac{1}{\sqrt{2\pi}} \exp \frac{\mathrm{x}^2}{2}, \ \Phi(\mathrm{x}) = \frac{1}{2}(1 + \text{ERF}(\mathrm{x})), \tag{8}$$

where $l$ is the lower bound, $u$ is the upper bound, $\mu$ is the pre-truncation mean, $\sigma$ is the pre-truncation standard deviation, and ERF is the error function. This distribution has the following NLL loss:

$$-\log \mathcal{TN}(\mathrm{x}|\mu, \sigma, l, u) = \frac{(\mathrm{x} - \mu)^2}{2\sigma^2} + \log \sqrt{2\pi\sigma^2} + \log \left( \Phi\left(\frac{u - \mu}{\sigma}\right) - \Phi\left(\frac{l - \mu}{\sigma}\right) \right) \tag{9}$$

Modeling $h^*$ as a $\mathcal{TN}$ instead of $\mathcal{N}$ presents several advantages. Firstly, $\mathcal{TN}$ constraints heuristic predictions to lie in the range $(l, u)$ given by the bounds of the distribution. Secondly, $\mathcal{TN}$ generalizes $\mathcal{N}$ as $\mathcal{TN}(\mu, \sigma, -\infty, \infty) = \mathcal{N}(\mu, \sigma)$ when no bounds are provided. Finally, $\mathcal{TN}$ opens the possibility for a variety of training scenarios for heuristic learning, with a sensible interpretation of each type of data, including the satisficing solution costs.

In this work, we focus on the scenario where an admissible heuristic is provided along with the optimal solution cost $h^*$ for each state, leaving other settings for future work. In this case, the admissible heuristic acts as the lower bound $l$ of $h^*$, which is modeled as a $\mathcal{TN}(\mathrm{x} = h^*|\mu, \sigma, l, \infty)$, where $\mu$ and $\sigma$ are predicted by an ML model. This model is trained by minimizing the NLL loss associated with this distribution. Note that we cannot model $h^*$ as $\mathcal{TN}(h^*|\mu, \sigma, h^*, h^*)$ since, during evaluation/test time, we do not have access to the optimal cost $h^*$. Also, this modeling decision is feasible even when no admissible heuristic is available (e.g., when the PDDL description of the environment is not known, as in Atari games (Bellemare et al., 2013)) since we can always resort to the blind heuristic $h^{\text{blind}}(s)$ or simply do $l = 0$, which still results in a tighter bound than the one provided by an untruncated Gaussian $\mathcal{N}(\mu, \sigma) = \mathcal{TN}(\mu, \sigma, -\infty, \infty)$.

Finally, our setting is orthogonal and compatible with *residual learning* (Yoon et al., 2008), where the ML model does not directly predict $\mu$ but rather predicts a *residual* or offset $\Delta\mu$ over a heuristic $h$, where $\mu = h + \Delta\mu$. Residual learning can be seen as initializing the model output $\mu$ around $h$, which, when $h$ is a good *unbiased estimator* of $h^*$, facilitates learning. This technique can be used regardless of whether $h^*$ is modeled as a $\mathcal{TN}$ or $\mathcal{N}$ because it merely corresponds to a particular implementation of $\mu = \mu_\theta(s)$, which is used by both distributions. It is an essential implementation detail because it is equivalent to the data normalization commonly applied to the standard regression tasks (rescaling and shifting the dataset to have mean 0, variance 1). However, the standard data normalization is not appropriate for heuristic learning and is inferior to residual learning because the target data is skewed above 0 and because the heuristic functions used as the basis of the residual is able to handle unknown, out-of-distribution data due to its symbolic nature.

## 4.1 Planning with a Truncated Gaussian

At planning time, we must obtain a point estimate of the output distribution, which will be used as a heuristic to determine the ordering between search nodes. As a point estimate, we can use any statistic of central tendency, thus we choose the mean. It is important to note that the $\mu$ parameter of $\mathcal{TN}(\mu, \sigma, l, u)$ is *not* the mean of this distribution since $\mu$ corresponds to the mean of $\mathcal{N}(\mu, \sigma)$ (i.e., the mean of the distribution *before truncation*) and does not necessarily lie in the interval $(l, u)$. The mean of a Truncated Gaussian is obtained according to Eq. 10. Note that a naive implementation of this formula results in rounding errors (See Appendix A for a numerically stable implementation).

$$\mathbb{E}[\mathrm{x}] = \mu + \sigma \frac{\phi(\frac{l-\mu}{\sigma}) - \phi(\frac{u-\mu}{\sigma})}{\Phi(\frac{u-\mu}{\sigma}) - \Phi(\frac{l-\mu}{\sigma})} \tag{10}$$

Eq. 10 satisfies $l \leq \mathbb{E}[x] \leq u$. This means that, when a lower bound $l$ is provided (e.g., by an admissible heuristic), the heuristic prediction returned by the model will never be smaller than $l$. Analogously, when an upper bound $u$ is also provided (e.g., by a satisficing solution cost), the model will never predict a heuristic value larger than $u$. With this, we hope that the use of a $\mathcal{TN}$ during planning helps the model make predictions that are closer to $h^*$ than the bounds themselves, potentially helping it achieve a super-symbolic improvement over admissible heuristics.

In contrast, the mode $\arg\max_x p(x)$ of $\mathcal{TN}$ is uninteresting: While we could use it as another point estimate, it is the same as the untruncated mean $\mu$ when the predicted $\mu$ is within the bounds, and equal to one of the upper/lower bounds otherwise. However, this inspires a naive alternative that is applicable even to $\mathcal{N}$, which is to clip the heuristic prediction $\mathbb{E}[x]$ (equal to $\mu$ for $\mathcal{N}$) to the interval $[l, u]$. We expect only a marginal gain from this trick because it only improves *really bad* predictions, i.e., those which would lie outside $[l, u]$ otherwise, and does not affect predictions that correctly lie inside $[l, u]$. In our experiments, we show that this approach is inferior to our first method.

Finally, we note that despite the use of admissible heuristics during training the learned heuristic is itself inadmissible. Thus, the natural way to evaluate it is to apply it to the satisficing (i.e., non-optimal) planning setting. In case a distributional estimate is used, i.e., when the ML model also learns to predict $\sigma$, we could discuss *likely-admissibility* (Ernandes & Gori, 2004; Marom & Rosman, 2020). However, this extension is left for future work.

## 5 EXPERIMENTAL EVALUATION

We evaluate the effectiveness of our new loss function under the domain-specific generalization setting, where the learned heuristic function is required to generalize across different problems of a single domain. Due to space limitations, we focus on the high-level descriptions and describe the detailed configurations in Appendix B.

**Data Generation.** We trained our system on three classical planning domains: blocksworld-4ops, logistics, and satellite. For each domain, we generated three sets of problem instances (train, validation, test) with parameterized generators used in past IPCs (Fawcett et al., 2011). We provided between 456 and 1536 instances for training (the variation is due to the difference in the number of generator parameters in each domain), between 132 and 384 instances for validation and testing (as separate sets), and 100 instances sampled from the test set for planning. Training, validation, and test instances are generated with the same range of generator parameters. Appendix describes the domains and parameter values employed. To generate the dataset from these instances, we optimally solved each instance with $A^*+h^{\text{LMcut}}$ in Fast Downward (Helmert, 2006) under 5 minutes and 8GB memory. When it failed to solve the instance within the limits, we discard it and retry generating and solving a new instance with a different random seed until it succeeds, which guarantees that it generates an specified number of instances. We also discard trivial instances that satisfy the goal conditions at the initial state. For each state $s$ in the optimal plan, we archived $h^*$ and the values of several heuristics (e.g., $h^{\text{LMcut}}$ and $h^{\text{FF}}$). This implies that each instance results in a different number of data points.

**Model Configurations.** We evaluated three neural network architectures to show that our statistical model is implementation agnostic. Neural Logic Machines (NLMs) (Dong et al., 2019) is an architecture designed for inductive learning and reasoning over symbolic data, which has been successfully applied to classical planning domains for learning heuristic functions (Gehring et al., 2022) with Reinforcement Learning (Sutton & Barto, 2018). STRIPS-HGN (Shen et al., 2020, HGN for short) is another architecture based on the notion of *hypergraphs*. Lastly, we included a linear regression with a hand-crafted feature set proposed in Gomoluch et al. (2017), which comprise the values of the goal-count (Fikes et al., 1972) and FF (Hoffmann & Nebel, 2001) heuristics, along with the total and mean number of effects ignored by FF's relaxed plan.

We analyze our learning & planning system from several orthogonal axes:

- **Truncated vs. Gaussian.** Using $\mu(s)$ as the parameter of a Gaussian $\mathcal{N}(\mu(s), \sigma(s))$ or Truncated Gaussian $\mathcal{TN}(\mu(s), \sigma(s), l, \infty)$ distribution.
- **Learned vs. fixed sigma.** Predicting $\sigma(s)$ with the model or using a constant value $\sigma(s) = \frac{1}{\sqrt{2}}$, as it is done for the MSE loss.

- **Lower bounds.** Computing the lower bound $l$ with the $h^{\mathrm{LMcut}}$ (Helmert & Domshlak, 2009) heuristic. When we employ a Gaussian distribution, $l$ is used to clip the heuristic prediction $\mathbb{E}[\mathrm{x}] = \mu(s)$ to the interval $[l, \infty)$. Ablation studies with $l = h^{\max}(s)$ (Bonet & Geffner, 2001) and $l = h^{\mathrm{blind}}(s)$ are included in the Appendix E.

- **Residual learning.** Either using the model to directly predict $\mu(s)$ or to predict an offset $\Delta\mu(s)$ over a heuristic $h(s)$, so that $\mu(s) = \Delta\mu(s) + h(s)$. We use $h = h^{\mathrm{FF}}$ as our unbiased estimator of $h^*$, as proposed in Yoon et al. (2008).

We trained each configuration with 5 different random seeds on a training dataset that consists of 400 problem instances subsampled from the entire training problem set (456-1536 instances, depending on the domain). Due to the nature of the dataset, these 400 problem instances can result in a different number of data points depending on the length of the optimal plan of each instance, but we do not control the final size of the data points. We performed $10^4$ weight updates (training steps) using *Adam* (Kingma & Ba, 2014) with batch size 256, learning rate of 0.01 for the linear regression and NLM, and 0.001 for HGN. All models use the NLL loss for training, motivated by the theory, but note that the NLL of $\mathcal{N}$ and fixed $\sigma = 1/\sqrt{2}$ matches the MSE up to a constant, as previously noted. For each model, we saved the weights that resulted in the best validation NLL loss during the training.

We report four different metrics computed on the test set: "NLL" (equivalent to the NLL loss), "MSE", "NLL+clip" and "MSE+clip". Here, "MSE" is a square error between $h^*$ and the point estimate used as the heuristic, i.e., for $\mathcal{TN}$ it is $(\mathbb{E}[x] - h^*)^2$, while for $\mathcal{N}$ it is simply $(\mu - h^*)^2$. "+clip" variants are exclusive to $\mathcal{N}$ and they clip $\mu$ to $l$, i.e., use $\max(\mu, l)$ in place of $\mu$ to compute the NLL and MSE. As a result, each configuration has 2x3 metrics to evaluate.

After the training, we use the point estimate provided by each model as a heuristic function for Greedy Best-First Search (GBFS) (Bonet & Geffner, 1999) to solve the set of 100 planning instances. To eliminate the effect of hardware (GPU and CPU) and software stack (e.g., deep learning library) differences, we evaluated the search performance using the number of heuristic evaluations required to solve each instance, rather than the total runtime. For each problem, we limited the number of evaluations to 10000. Additionally, we evaluated GBFS with off-the-shelf $h^{\mathrm{FF}}$ heuristic as a baseline. The planning component is based on Pyperplan (Alkhazraji et al., 2020).

## 5.1 Training Accuracy Evaluations

We focus on the results obtained by the NLM models, as the results obtained by the Linear and the HGN models (Appendices C and D) were not substantially different. Table 1 shows the test metrics obtained by the different configurations. $\mathcal{TN}$ obtains around 40% lower MSE than $\mathcal{N}$+clip on (geometric) average. In addition, regardless of the evaluated metric (NLL or MSE), $\mathcal{TN}$ outperforms $\mathcal{N}$ for all configurations except *fixed/$h^{\mathrm{FF}}$* on logistics, where it obtains a slightly worse MSE (but much better NLL nonetheless). In contrast, $\mathcal{N}$+clip only marginally improves over $\mathcal{N}$, confirming our hypothesis that post-hoc clipping is insufficient for an accurate prediction.

Additional detailed observations follows. Firstly, we obtained the mean square error $(h^{\mathrm{FF}} - h^*)^2$ between $h^{\mathrm{FF}}$ and $h^*$ and observed that the trained heuristics, including those that use residual learning from $h^{\mathrm{FF}}$, tend to be more accurate than $h^{\mathrm{FF}}$ (see third table column). Secondly, residual learning often improves performance considerably, thus proving to be an effective way of utilizing inadmissible heuristics. Finally, the $\mathcal{TN}$ tends to converge faster during training, as shown in Fig. 2.

## 5.2 Search Performance Evaluations

We compared the search performance of GBFS using heuristic functions obtained by different models, which include our proposed configuration *learn/$h^{\mathrm{FF}}$* (best accuracy in Table 1) and the baseline *fixed/none* configuration which follows the common practice in the literature using a fixed $\sigma = 1/\sqrt{2}$ and lacking residual learning.

Table 2 shows the average±stdev of the number of problems solved (the *coverage* metric) and the average number of node evaluations per problem over 5 seeds. The second metric is necessary because the coverage can saturate to 100 and becomes statistically insignificant when all/most problem instances are solved. The accuracy in Table 1 strongly correlates with the search performance in Table

Table 1: Test metrics for NLM. For each experiment configuration, we performed $10^4$ training steps, saving the checkpoints with the best validation NLL loss. We tested several orthogonal configurations: 1) Learning $\sigma$ (*learn*) or fixing it to $1/\sqrt{2}$ (*fixed*) and 2) Using residual learning ($h^{\text{FF}}$) or not (*none*). For each configuration, we compare the test NLL and MSE of the Gaussian ($\mathcal{N}$) and Truncated Gaussian ($\mathcal{TN}$) models. Rows labeled as *+clip* denote a $\mathcal{N}$ model where $\mu$ is clipped above $h^{\text{LMcut}}$. For each configuration, the best metric among $\mathcal{N}$, $\mathcal{N}$+clip, and $\mathcal{TN}$ is highlighted in **bold**. Results from linear regression and STRIPS-HGN are provided in the Appendix.

| domain | metric | learn/$h^{\text{FF}}$ | | learn/none | | fixed/$h^{\text{FF}}$ | | fixed/none | | $h^{\text{FF}}$ |
|---|---|---|---|---|---|---|---|---|---|---|
| | | $\mathcal{N}$ | $\mathcal{TN}$ | $\mathcal{N}$ | $\mathcal{TN}$ | $\mathcal{N}$ | $\mathcal{TN}$ | $\mathcal{N}$ | $\mathcal{TN}$ | |
| blocks-4ops | NLL | .16 | **.02** | .85 | **.54** | 1.37 | **.30** | 1.49 | **.44** | |
| | +clip | .15 | | .81 | | 1.37 | | 1.48 | | |
| | MSE | .67 | **.55** | 1.48 | **1.21** | .43 | **.41** | .91 | **.73** | 16.61 |
| | +clip | .67 | | 1.46 | | .42 | | .88 | | 16.37 |
| logistics | NLL | .32 | **.09** | 1.49 | **.67** | 1.33 | **.34** | 2.24 | **.91** | |
| | +clip | .31 | | 1.33 | | 1.32 | | 1.97 | | |
| | MSE | .39 | **.30** | 4.72 | **.44** | .25 | .28 | 3.92 | **1.34** | .78 |
| | +clip | .38 | | 3.22 | | **.24** | | 2.81 | | .78 |
| satellite | NLL | -.13 | **-.48** | 1.00 | **-.05** | 1.41 | **-.11** | 1.66 | **-.07** | |
| | +clip | -.15 | | .77 | | 1.38 | | 1.51 | | |
| | MSE | .80 | **.34** | 1.93 | **.41** | .56 | **.39** | 1.59 | **.48** | .92 |
| | +clip | .71 | | 1.09 | | .45 | | .97 | | .92 |

Table 2: Planning results on NLM weights saved according to the best validation NLL loss, comparing the average±stdev of the number of instances solved under $10^4$ node evaluations and the average number of evaluated nodes across the problems. The number of evaluated nodes is counted as $10^4$ on instances that the planner failed to solve. For each configuration, the best metric among $\mathcal{N}$, $\mathcal{N}$+clip, and $\mathcal{TN}$ is highlighted in **bold**, with statistically insignificant ties being equally highlighted.

| domain | $h^{\text{FF}}$ | learn/$h^{\text{FF}}$ | | | fixed/none | | |
|---|---|---|---|---|---|---|---|
| | | $\mathcal{N}$ | $\mathcal{N}$+clip | $\mathcal{TN}$ | $\mathcal{N}$ | $\mathcal{N}$+clip | $\mathcal{TN}$ |
| Number of solved instances under $10^4$ evaluations (higher the better) | | | | | | | |
| blocks-4opts | 55 | **97.4±1.14** | **97.4±1.14** | **98±1.87** | **98.6±1.14** | 92±3.32 | 90.2±3.11 |
| logistics | 99 | 93±5.83 | **97.6±1.67** | **97.4±2.70** | 40.4±5.55 | 82.2±5.22 | **90.6±3.91** |
| satellite | 84 | 34.4±7.27 | 55.8±9.26 | **65.2±3.27** | 16.2±4.76 | 48±4.06 | **62.6±8.71** |
| Average node evaluations (smaller the better) | | | | | | | |
| blocks-4opts | 5751 | 656 | 656 | **560** | **640** | 1537 | 1514 |
| logistics | 1031 | 1435 | 1077 | **953** | 6841 | 2986 | **1830** |
| satellite | 3398 | 7227 | 5557 | **4594** | 8792 | 6336 | **4862** |

2 as *learn/$h^{\text{FF}}$* tends to outperform *fixed/none* and $\mathcal{TN}$ tends to outperform $\mathcal{N}$ and $\mathcal{N}$+clip. Note that the coverage in blocksworld tends to saturate, thus the improvements are in the node evaluations.

$\mathcal{TN}$ did not outperform $\mathcal{N}$ in *fixed/none* in blocksworld but this is expected: *fixed/none* is known to be a problematic configuration because it contains an ad-hoc assumption on $\sigma$ and lacks the normalization by the residual base, both of which could prevent it from learning a useful search guidance. We still maintain our conclusion because the best *learn/$h^{\text{FF}}$* blocksworld model ($\mathcal{TN}$, 540 evaluations) outperforms the best *fixed/none* blocksworld model ($\mathcal{N}$, 640 evaluations).

Compared to the State-of-the-Art off-the-shelf heuristic $h^{\text{FF}}$, our models significantly outperform $h^{\text{FF}}$ in blocksworld, although $h^{\text{FF}}$ is competitive in logistics and dominates our models in satellite. This is surprising because the $\mathcal{TN}$ *learn/$h^{\text{FF}}$* models are significantly more accurate (in MSE) than

$h^{\mathrm{FF}}$ in every domain. Our hypothesis is that the GBFS algorithm used for evaluation may be sensitive to the small changes in node ordering, which matters more (Garrett et al., 2016) especially with accurate heuristics ($h^{\mathrm{FF}}$ is already quite accurate in logistics and satellite with MSE below 1) and can produce unreliable results. Investigating the theoretical characteristics of GBFS (Heusner et al., 2018; Kuroiwa & Beck, 2022) is an important future work, but is outside the scope of this paper.

## 6    RELATED WORK

ML approaches for learning search guidance in Classical Planning can be categorized according to two orthogonal dimensions: Supervised Learning (SL) vs. Reinforcement Learning (RL), and policy learning (i.e., predicting the next action) vs. heuristic learning (i.e., predicting a cost/reward as a heuristic/value function). SL approaches (Yoon et al., 2006; 2008; Arfaee et al., 2011; Satzger & Kramer, 2013; Gomoluch et al., 2017; Shen et al., 2020) work on a pre-generated dataset, which may be a disadvantage if obtaining a high-quality dataset (i.e., with optimal costs) is costly. SL can also be applied to policy learning (Toyer et al., 2018). On the other hand, RL integrates the data-collection process into the training by allowing the agent to interact with the environment, with the additional benefit that it is compatible with model-free settings (Mnih et al., 2015; Silver et al., 2016). RL has been successfully applied to classical planning in the context of both policy learning (Rivlin et al., 2019) and heuristic learning (Gehring et al., 2022). Its main drawback is the high variance/instability of training (Henderson et al., 2018) and its sample inefficiency (Badia et al., 2020).

Other works explore orthogonal ideas such as learning residuals/offsets from existing heuristics (Ernandes & Gori, 2004; Yoon et al., 2008; Satzger & Kramer, 2013; Gehring et al., 2022), learning to rank states (Garrett et al., 2016), learning pruning rules (Krajňanský et al., 2014), improving the sampling and data generation process (Arfaee et al., 2011; Ferber et al., 2022) and learning from regression (Yu et al., 2020).

## 7    CONCLUSION AND FUTURE WORK

In this paper, we studied the problem of supervised heuristic learning under a statistical lens, focusing on how to effectively utilize the information provided by admissible heuristics. Firstly, we provided some statistical background on heuristic learning which was later leveraged to analyze the decisions made (sometimes unknowingly) in the literature. We explained how the commonly used MSE loss implicitly models the heuristic to be learned as a Gaussian distribution. Then, we argued that this heuristic should instead be modeled as a Truncated Gaussian, where admissible heuristics are used as the lower bound of the distribution. We conducted extensive experimentation, comparing the heuristics learned with our truncated-based statistical model versus those learned by minimizing squared errors. Results show that our proposed method improves convergence speed during training and yields better heuristics (with 40% lower MSE on average), thus confirming that it is the correct approach for utilizing admissible bounds in heuristic learning.

Our findings serve to answer the three important questions we raised in the introduction: **What should be learned in heuristic learning?** We should use $h^*$ as the training target, instead of learning from admissible heuristics or sub-optimal plan costs, since otherwise no super-symbolic benefit can be achieved. **How should we learn?** We should follow the MLE approach, modeling $h^*$ as a Truncated Gaussian whose lower bound is given by an admissible heuristic. **Why so?** According to the principle of maximum entropy, a Truncated Gaussian is the distribution that encodes all our prior knowledge without introducing any extra assumption into the model that may result in harmful bias.

In future work, we will extend our proposed method to other learning settings. One interesting scenario is given by iterative search algorithms (Richter et al., 2010; 2011), where the cost of the best solution found so far could be used as the upper bound of a Truncated Gaussian. We are also interested in discovering whether our method can be successfully applied to learn a heuristic when optimal costs are not available, but only their upper and/or lower bounds, using *variational inference* (Jordan et al., 1999; Kingma & Welling, 2013) with the optimal cost as a hidden variable. We also plan to explore the Reinforcement Learning setting where a value function is learned instead of a heuristic, extending the work on residual learning for RL (Gehring et al., 2022). Finally, the statistical framework for heuristic learning extends beyond classical planning. Application to logistics-type industrial domains (e.g., TSP, VRP) is an interesting avenue for future work.

## 8 Reproducibility Statement

In order to support reproducibility, we provide all the code used in this work as part of the supplementary materials. This includes the code for generating the datasets, training and testing the models, and running the planning experiments. The particular datasets the experiments were performed on are also included in the supplementary materials.

In Appendix A, we explain how the Truncated Gaussian distribution used in this work was implemented. The Pytorch code of this implementation can be found along with the rest of the code in the supplementary materials.

Finally, Section 5 provides details about the experimental setup, e.g., the size of the datasets employed and values for training parameters such as the learning rate. The remaining details can be found in the Appendices. Appendix B.1 contains the hyperparameter values for the different ML models (NLM, STRIPS-HGN and Linear Regression), whereas Appendix B.2 contains the parameter values used as inputs to the instance generators, in order to generate the problems for the different domains.

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

# A  TRUNCATED GAUSSIAN IMPLEMENTATION

This Appendix explains several important implementation details of the Truncated Gaussian distribution used in our work. Our Pytorch implementation can be found in the supplementary material, along with the rest of the code.

## A.1  NUMERICALLY STABLE FORMULAS FOR TRUNCATED GAUSSIAN

The Truncated Gaussian distribution is a four-parameter probability distribution defined as follows:

$$\mathcal{N}(\mathrm{x}|\mu, \sigma, l, u) = \begin{cases} \frac{1}{\sigma} \frac{\phi(\frac{\mathrm{x}-\mu}{\sigma})}{\Phi(\frac{u-\mu}{\sigma}) - \Phi(\frac{l-\mu}{\sigma})} & l \leq \mathrm{x} \leq u \\ 0 & \text{otherwise.} \end{cases}$$

$$\text{where } \phi(\mathrm{x}) = \frac{1}{\sqrt{2\pi}} \exp \frac{\mathrm{x}^2}{2},$$

$$\Phi(\mathrm{x}) = \frac{1}{2}(1 + \mathrm{ERF}(\mathrm{x})).$$

In order to train and use a system that involves a Truncated Gaussian, we need to compute several properties, such as its mean and the log-probability of some value $x$ under the distribution. However, the naive implementation of the formulas for calculating these quantities is numerically unstable due to floating-point rounding errors, especially when $\mu$ lies outside the interval $(l, u)$. In this subsection, we briefly explain the source of instability and provide numerically stable formulas for calculating these values.

Given a Truncated Gaussian distribution $\mathcal{N}(\mathrm{x} \mid \mu, \sigma, l, u)$, its mean $\mathbb{E}[\mathrm{x}]$ is given by the following formula:

$$\mathbb{E}[\mathrm{x}] = \mu + \frac{\phi(\alpha) - \phi(\beta)}{\Phi(\beta) - \Phi(\alpha)}\sigma, \quad \text{where}$$

$$\alpha = \tfrac{l-\mu}{\sigma}, \quad \beta = \tfrac{u-\mu}{\sigma}. \quad (\beta \geq \alpha)$$

The expression $\frac{\phi(\alpha)-\phi(\beta)}{\Phi(\beta)-\Phi(\alpha)}$ should not be evaluated directly because it involves subtractions between values that could be potentially very close to each other, causing floating-point rounding errors.

We now describe a stable implementation of this formula introduced by Fernandez-de Cossio-Diaz (2018). Let us define the following function:

$$F_1(x, y) = \frac{e^{-x^2} - e^{-y^2}}{\mathrm{ERF}(y) - \mathrm{ERF}(x)}$$

Then, we reformulate the mean as follows:

$$\mathbb{E}[\mathrm{x}] = \mu + \sqrt{\frac{2}{\pi}} F_1\left(\frac{\alpha}{\sqrt{2}}, \frac{\beta}{\sqrt{2}}\right)\sigma.$$

$F_1$ can be evaluated in a numerically stable manner by using the formulas below:

$$\begin{aligned}
&F_1(x, y) \\
&= F_1(y, x), && \text{if } |x| > |y| \\
&= P_1(x, y - x), && \text{if } |x - y| = |\epsilon| < 10^{-7} \\
&= \frac{1 - \Delta}{\Delta \mathrm{ERFCX}(-y) - \mathrm{ERFCX}(-x)}, && \text{if } x, y \leq 0 \\
&= \frac{1 - \Delta}{\mathrm{ERFCX}(x) - \Delta \mathrm{ERFCX}(y)}, && \text{if } x, y \geq 0 \\
&= \frac{(1 - \Delta)e^{-x^2}}{\mathrm{ERF}(y) - \mathrm{ERF}(x)}, && \text{otherwise.}
\end{aligned}$$

where $\Delta = e^{x^2 - y^2}$, $\text{ERFCX}(x) = e^{x^2} \text{ERFC}(x)$ is a function that is commonly available in mathematical packages, and $P_1$ is a Taylor expansion of $F_1(x, x + \epsilon) = P_1(x, \epsilon)$ where $y = x + \epsilon$:

$$P_1(x, \epsilon) = \sqrt{\pi} x + \frac{1}{2} \sqrt{\pi} \epsilon - \frac{1}{6} \sqrt{\pi} x \epsilon^2 - \frac{1}{12} \sqrt{\pi} \epsilon^3 +$$
$$\frac{1}{90} \sqrt{\pi} x (x^2 + 1) \epsilon^4.$$

Next, we provide a numerically stable method for computing the log-probability $\log \mathcal{N}(x \mid \mu, \sigma, l, u)$. Let us assume $l \leq x \leq u$, since otherwise the probability is 0 (whose logarithm is $-\infty$). The value is given by the following expression:

$$\log \mathcal{N}(x \mid \mu, \sigma, l, u) = \log \left( \frac{1}{\sigma} \frac{\phi(\xi)}{\Phi(\beta) - \Phi(\alpha)} \right) =$$
$$- \log \sigma - \log \sqrt{2\pi} - \frac{\xi^2}{2} - \log \big( \Phi(\beta) - \Phi(\alpha) \big),$$
$$\text{where} \quad \xi = \frac{x - \mu}{\sigma}.$$

Let $Z = \Phi(\beta) - \Phi(\alpha)$. We obtain $\log(Z)$ from the stable formula for $\mathbb{E}[\text{x}]$. When $\alpha, \beta \geq 0$,

$$\log(Z) = - \log \frac{\mathbb{E}[\text{x}] - \mu}{\sigma} - \log \sqrt{2\pi} - \frac{\alpha^2}{2} +$$
$$\log \left( 1 - e^{\frac{\alpha^2 - \beta^2}{2}} \right).$$

When $\alpha, \beta \leq 0$,

$$\log(Z) = - \log \frac{\mu - \mathbb{E}[\text{x}]}{\sigma} - \log \sqrt{2\pi} - \frac{\beta^2}{2} +$$
$$\log \left( 1 - e^{\frac{\beta^2 - \alpha^2}{2}} \right).$$

Otherwise,

$$\log(Z) = - \log 2 + \log \left[ \text{ERF} \left( \frac{\beta}{\sqrt{2}} \right) - \text{ERF} \left( \frac{\alpha}{\sqrt{2}} \right) \right].$$

### A.2  TRUNCATED GAUSSIAN WITH MISSING BOUNDS

A Truncated Gaussian distribution can be defined with either the lower $l$ or upper bound $u$ missing, as $\mathcal{N}(\mu, \sigma, -\infty, u)$ or $\mathcal{N}(\mu, \sigma, l, \infty)$, respectively. It can also be defined with no bounds at all as $\mathcal{N}(\mu, \sigma, -\infty, \infty)$, in which case it is equivalent to an untruncated Gaussian $\mathcal{N}(\mu, \sigma)$.

In our implementation, we use $l = -1e5$ and $u = 1e5$ as the parameters of a Truncated Gaussian with no lower and/or upper bound, respectively. We have observed that these values result indistinguishable from $l \to -\infty$ and $u \to \infty$ when calculating the mean $\mathbb{E}[\text{x}]$ and log-probability $\log p(x)$, as long as $-1e5 \ll \mu \ll 1e5$, $\sigma \ll 1e5$ and $-1e5 < x < 1e5$ (since $p(x) = 0$ for any $x$ outside the interval $[l, u]$).

### A.3  TRUNCATED GAUSSIAN WITH OPEN BOUNDS

When defining a Truncated Gaussian distribution $\mathcal{N}(\mu, \sigma, l, u)$, we need to specify whether the bounds $l, u$ are contained in the support of the distribution or not, i.e., whether the support is equal to $[l, u]$ (they are contained) or $(l, u)$ (they are *not* contained). When the support is $[l, u]$ we say that the Truncated Gaussian has *closed bounds* and that it has *open bounds* otherwise.

Our first Truncated Gaussian implementation used closed bounds, but we discovered that this decision would sometimes lead to learning issues since the ML model would tend to output $\mu \ll 0$ (e.g., $\mu = -100$). We believe the reason for that behavior is that a highly accurate lower bound

$l$ (e.g., $h^{\text{LMcut}}$) can be sometimes equal to $h^*$ and the ML model is encouraged to maximize $\log p(l) = \log p(h^*)$. In order to do so, it can simply output $\mu \ll 0$, as the smaller (more negative) $\mu$ gets, the higher $\log p(l)$ becomes. Therefore, using closed bounds would often result in a learned heuristic equivalent to $l = h^{\text{LMcut}}$, as the mean of $\mathcal{N}(\mu, \sigma, l, u)$ is almost equal to $l$ when $\mu \ll l$.

For this reason, we switched to open bounds in our implementation. To do so, we simply subtracted a small value $\epsilon = 0.1$ from $l$, obtaining a new distribution $\mathcal{N}(\mu, \sigma, l - \epsilon, u)$. This made sure that $x$ was never equal to $l' = l - \epsilon$ when calculating $\log p(x = h^*)$, which prevented the ML model from predicting $\mu \ll 0$. Finally, in order to obtain a Truncated Gaussian where the upper bound $u$ is also *open*, we can add $\epsilon$ to $u$, which results in a new distribution $\mathcal{N}(\mu, \sigma, l - \epsilon, u + \epsilon)$.

# B  PARAMETER DETAILS

## B.1  MODEL HYPERPARAMETERS

In this Appendix, we detail the hyperparameter values used for the different models: NLM, HGN, and linear regression. In general, we did not perform extensive hyperparameter tuning for the different models.

For the NLM, we used a model with breadth 3 and depth 5, where every inner layer outputs 8 predicates for each arity. The multi-layer perceptrons used in the network employed sigmoid as their activation function and contained no hidden layer.

For the HGN, we employed a *hidden size* of 32 and 3 recursion steps. We note that using more recursion steps did not improve performance significantly (but incurred in higher computational cost) and, in satellite, using 4 recursion steps actually degraded performance. As mentioned in the main paper, the learning rate for the HGN is $1e^{-3}$, which corresponds to the value used in Shen et al. (2020).

Finally, we report that we initially tested an L2 weight decay penalty for the linear regression model but removed it because it did not help the model.

## B.2  PARAMETERS OF INSTANCE GENERATORS

As mentioned in the paper, we generated the problems used in our experiments with parameterized generators (Fawcett et al., 2011). For each domain, we tried to select a diverse set of parameters that resulted in problems solvable in under 5 minutes and 8GB of memory using $A^* + h^{\text{LMcut}}$ in Fast Downward. We now detail the range of parameter values used for each generator. For each parameter combination, we generated one or several problems, discarding those resulting from invalid combinations. In blocksworld, we generated problems containing between 5 and 15 blocks. In logistics, we used the following set of parameter values: $airplanes = 1..3$, $cities = 1..5$, $city\ size = 1..3$, $packages = 3..5$, $trucks = 5$. In satellite, we utilized the following: $satellites = 1..5$, $max\ instruments\ per\ satellite = 3$, $modes = 3..5$, $targets = [7, 10, 15]$, $observations = 3..5$.

# C EXPERIMENTAL RESULTS FOR LINEAR REGRESSION

In this Appendix, we provide the results of the experiments conducted on the Linear Regression (LR) models. These experiments were run using the same parameters as in the NLM experiments.

Table S3 shows the NLL and MSE metrics obtained by the LR model on the test problems. The trend obtained by the LR model is very similar to that of the NLM model: $\mathcal{TN}$ obtains better test MSE and NLL than $\mathcal{N}$ for almost every configuration. In those few cases where $\mathcal{N}$ outperforms $\mathcal{TN}$, it only does so slightly. Additionally, we observe that $\mathcal{N} + clip$ only provides a marginal improvement over $\mathcal{N}$ and that residual learning provides no benefit at all. Residual learning does not help the LR model because it already receives the $h^{\mathrm{FF}}$ heuristic as one of its inputs, so using $h^{\mathrm{FF}}$ again as the basis for the residual does not provide any additional information.

We note that the LR models are remarkably accurate in logistics and satellite compared to the NLM ones despite their simplicity. Our hypothesis is that since $h^{\mathrm{FF}}$ is already informative in those domains, it is easy for the LR models to obtain good results. On the other hand, the accuracy of LR is not as good in blocksworld because $h^{\mathrm{FF}}$ tends to have large errors and, therefore, the $h^{\mathrm{FF}}$ input (as well as the residual) is not helpful. Nonetheless, the LR models manage to outperform $h^{\mathrm{FF}}$ in blocksworld by a great margin in terms of accuracy.

Table S4 shows the results of the planning experiments for the LR models. We also observe that residual learning does not improve performance (since LR models already receive $h^{\mathrm{FF}}$ as an input). The trends are similar to the NLM results overall: The LR models outperform $h^{\mathrm{FF}}$ in blocksworld, obtain similar results in logistics and perform worse in satellite.

Table S3: Test metrics for Linear Regression (LR) model. Table columns and rows have the same meaning as in Table 1 of the main paper. For each configuration, the best metric among $\mathcal{N}$, $\mathcal{N}+clip$, $\mathcal{TN}$ is highlighted in **bold**. $\mathcal{TN}$ improves the accuracy over $\mathcal{N}$ and $\mathcal{N}$+clip overall.

| domain | metric | Weights with the best validation NLL in $10^4$ steps | | | | | | | | $h^{\mathrm{FF}}$ |
| | | learn/$h^{\mathrm{FF}}$ | | learn/none | | fixed/$h^{\mathrm{FF}}$ | | fixed/none | | |
| | | $\mathcal{N}$ | $\mathcal{TN}$ | $\mathcal{N}$ | $\mathcal{TN}$ | $\mathcal{N}$ | $\mathcal{TN}$ | $\mathcal{N}$ | $\mathcal{TN}$ | |
| blocks-4ops | NLL | 1.92 | **1.30** | 1.92 | **1.30** | 2.53 | **2.01** | 2.53 | **2.01** | |
| | +clip | 1.89 | | 1.89 | | 2.50 | | 2.50 | | |
| | MSE | 6.02 | **5.17** | 6.08 | **5.19** | 5.06 | 5.08 | 5.06 | 5.09 | 16.61 |
| | +clip | 6.01 | | 6.07 | | **4.94** | | **4.94** | | 16.37 |
| logistics | NLL | .85 | **.68** | .89 | **.71** | 1.42 | **.71** | 1.44 | **.71** | |
| | +clip | .85 | | .88 | | 1.42 | | 1.43 | | |
| | MSE | .65 | **.52** | .68 | **.51** | .64 | .64 | .68 | **.66** | .78 |
| | +clip | .65 | | .68 | | **.63** | | .68 | | .78 |
| satellite | NLL | .69 | **.13** | .61 | **.13** | 1.50 | **.26** | 1.47 | **.24** | |
| | +clip | .69 | | .61 | | 1.48 | | 1.45 | | |
| | MSE | 1.05 | **.48** | .96 | **.49** | .92 | **.70** | .80 | **.63** | .92 |
| | +clip | 1.05 | | .96 | | .86 | | .74 | | .92 |

Table S4: Planning results for Linear Regression (LR) model. For each model, we use the weights that resulted in the best validation NLL loss during training. Table columns and rows have the same meaning as in Table 2 of the main paper. For each configuration, the best metric among $\mathcal{N}$, $\mathcal{N}$+clip and $\mathcal{TN}$ is highlighted in **bold**.

| domain | | learn/$h^{\mathrm{FF}}$ | | | fixed/none | | |
| | $h^{\mathrm{FF}}$ | $\mathcal{N}$ | $\mathcal{N}$+clip | $\mathcal{TN}$ | $\mathcal{N}$ | $\mathcal{N}$+clip | $\mathcal{TN}$ |
| --- | --- | --- | --- | --- | --- | --- | --- |
| | | Number of solved instances under $10^4$ evaluations | | | | | |
| blocksworld | 55 | **99.6±.55** | **99.6±.55** | 98.2±1.79 | 98±2.83 | 98±2.92 | 97.8±1.48 |
| logistics | 99 | 95.8±2.17 | 96±1.87 | 97.8±1.64 | 99.6±.55 | 100±.00 | 99.6±.89 |
| satellite | 84 | 33.4±3.21 | 53.2±5.93 | 59.4±11.41 | 68.4±8.23 | 75.2±4.38 | 65.2±8.11 |
| | | Average node evaluations | | | | | |
| blocksworld | 5753 | 4455 | **4350** | 4851 | **4733** | 4858 | 4815 |
| logistics | 1033 | 1094 | 1073 | **943** | 1040 | 1056 | **1018** |
| satellite | 3400 | **3412** | 3779 | 5501 | **3578** | 3642 | 4524 |

## D  EXPERIMENTAL RESULTS FOR HGNS

In this Appendix, we provide the results of the experiments conducted on the STRIPS-HGN models. HGN models are trained with learning rate $1e^{-3}$.

The test metrics obtained by the HGN model are shown in Table S5, whereas Table S6 shows its planning results. With HGN, $\mathcal{TN}$ tends to improve NLL and MSE over $\mathcal{N}$ and $\mathcal{N}$+clip, but less effectively so than with NLM (11/12 configurations for NLL, 8/12 configurations for MSE). One potential reason is the limited expressivity of HGN: Unlike NLM, HGN is designed to only receive the delete-relaxed information about the problem instance, which may harm its ability to learn a heuristic for the original instance. Another potential reason for its weak performance is training time: We observed that HGN requires a smaller learning rate to achieve training stability, which means that it may need more training steps in order to converge.

Focusing on the *learn/h*$^{\mathrm{FF}}$ configuration, the planning results in Table S6 show a general improvement of HGN over $h^{\mathrm{FF}}$ in blocksworld (both for coverage and number of node evaluations) and logistics (coverage is saturated, but node evaluations improves over $h^{\mathrm{FF}}$). Similarly to the NLM results, the learned heuristics did not outperform $h^{\mathrm{FF}}$ in satellite. Interestingly, the rankings (order) for the two metrics (coverage and node evaluations) are the opposite in satellite: $\mathcal{TN}$ results in better (fewer) evaluations than $\mathcal{N}$ while solving fewer instances. This may indicate overfitting to a particular subset of similar instances, which it solves quickly and thus contributes to fewer evaluations, while failing on other instances, which reduces coverage. This is also natural considering the limited expressivity of HGNs.

Table S5: Test metrics for HGN model. Table columns and rows have the same meaning as in Table 1 of the main paper. For each configuration, the best metric among $\mathcal{N}$, $\mathcal{N} + clip$, $\mathcal{TN}$ is highlighted in **bold**. $\mathcal{TN}$ tends to improve accuracy over $\mathcal{N}$ and $\mathcal{N}$+clip.

| domain | metric | learn/$h^{FF}$ $\mathcal{N}$ | $\mathcal{TN}$ | learn/none $\mathcal{N}$ | $\mathcal{TN}$ | fixed/$h^{FF}$ $\mathcal{N}$ | $\mathcal{TN}$ | fixed/none $\mathcal{N}$ | $\mathcal{TN}$ | $h^{FF}$ |
|---|---|---|---|---|---|---|---|---|---|---|
| blocks-4ops | NLL | **-.23** | -.18 | .33 | **.14** | 1.31 | **.24** | 1.36 | **.26** | |
| | +clip | **-.23** | | .30 | | 1.31 | | 1.36 | | |
| | MSE | **.32** | .37 | .72 | **.52** | **.16** | .20 | .39 | **.30** | 16.61 |
| | +clip | **.32** | | .72 | | **.16** | | .38 | | 16.37 |
| logistics | NLL | .56 | **.31** | .61 | **.46** | 1.34 | **.35** | 1.36 | **.37** | |
| | +clip | .56 | | .54 | | 1.33 | | 1.36 | | |
| | MSE | .49 | **.42** | .46 | .52 | **.28** | .29 | .39 | **.33** | .78 |
| | +clip | .49 | | **.43** | | **.28** | | .36 | | .78 |
| satellite | NLL | -.03 | **-.57** | .19 | **-.24** | 1.36 | **-.13** | 1.39 | **-.19** | |
| | +clip | -.06 | | .04 | | 1.35 | | 1.36 | | |
| | MSE | .64 | **.27** | .73 | **.27** | .39 | **.31** | .49 | **.26** | .92 |
| | +clip | .55 | | .49 | | .33 | | .36 | | .92 |

Table S6: Planning results for HGN model. For each model, we use the weights that resulted in the best validation NLL loss during training. Table columns and rows have the same meaning as in Table 2 of the main paper. For each configuration, the best metric among $\mathcal{N}$, $\mathcal{N}$+clip and $\mathcal{TN}$ is highlighted in **bold**.

| domain | $h^{FF}$ | learn/$h^{FF}$ $\mathcal{N}$ | $\mathcal{N}$+clip | $\mathcal{TN}$ | fixed/none $\mathcal{N}$ | $\mathcal{N}$+clip | $\mathcal{TN}$ |
|---|---|---|---|---|---|---|---|
| | | | Number of solved instances under $10^4$ evaluations | | | | |
| blocksworld | 55 | **70.2±.45** | **71.8±1.92** | 63.6±2.61 | **67.6±3.05** | 66.8±2.17 | 64.4±2.97 |
| logistics | 99 | 98.6±.89 | 98.8±.84 | **100±.00** | 99.2±.84 | 99±.71 | 99±.71 |
| satellite | 84 | **85±1.00** | 79.8±1.92 | 56.2±11.65 | **84±1.41** | 82.8±2.77 | 72.6±1.82 |
| | | | Average node evaluations | | | | |
| blocksworld | 5751 | 311 | **310** | 451 | **463** | 487 | 515 |
| logistics | 1031 | 1292 | 1190 | **982** | 748 | **660** | 724 |
| satellite | 3398 | 7064 | 5713 | **5042** | 4298 | **3871** | 4655 |

# E   EXPERIMENTAL RESULTS WITH DIFFERENT LOWER BOUNDS

Tables S7-S9 show the results obtained by the different models using $h^{\mathrm{max}}$ and $h^{\mathrm{blind}}$ as the lower bound $l$. The results obtained with $l = h^{\mathrm{max}}$ and $l = h^{\mathrm{blind}}$ are generally worse than those obtained with $l = h^{\mathrm{LMcut}}$. $\mathcal{TN}$ tends to perform better than $\mathcal{N}$ while there are little to no improvements in $\mathcal{N}$+clip over $\mathcal{N}$. In all cases, the satellite domain remains challenging.

We believe this behaviour is due to the quality of the lower bound $l$, or its lack thereof. The issue is especially significant in $\mathcal{N}$+clip because the clipping has no effect on the prediction unless $l$ is sufficiently large, while $\mathcal{TN}$ receives the adjustment to $\mu$ even when $l < \mu$. Nonetheless, the adjustment is small if $l \ll \mu$, as demonstrated by the lack of large improvements by $l = h^{\mathrm{blind}}$.

Table S7: Test metrics for the NLM models with $l = h^{\mathrm{max}}$ and $l = h^{\mathrm{blind}}$, using the *learn/h*$^{\mathrm{FF}}$ configuration. For each experiment configuration, we performed $10^4$ training steps, saving the checkpoints with the best validation NLL loss.

| domain | metric | Weights with best val NLL | | | |
| --- | --- | --- | --- | --- | --- |
| | | $h^{\mathrm{max}}$ | | $h^{\mathrm{blind}}$ | |
| | | $\mathcal{N}$ | $\mathcal{TN}$ | $\mathcal{N}$ | $\mathcal{TN}$ |
| blocks-4ops | NLL | .15 | **.06** | **.18** | **.18** |
| | +clip | .15 | | **.18** | |
| | MSE | **.65** | .65 | .69 | **.62** |
| | +clip | **.65** | | .69 | |
| logistics | NLL | .30 | **.28** | .30 | **.22** |
| | +clip | .30 | | .30 | |
| | MSE | **.39** | .42 | **.39** | **.39** |
| | +clip | **.39** | | **.39** | |
| satellite | NLL | **-.13** | -.04 | **-.13** | .17 |
| | +clip | **-.13** | | **-.13** | |
| | MSE | **.84** | .92 | **.84** | .90 |
| | +clip | **.84** | | **.84** | |

Table S8: Test metrics for the LR models with $l = h^{\mathrm{max}}$ and $l = h^{\mathrm{blind}}$, using the *learn/$h^{\mathrm{FF}}$* configuration. For each experiment configuration, we performed $10^4$ training steps, saving the checkpoints with the best validation NLL loss.

| domain | metric | Weights with best val NLL | | | |
| --- | --- | --- | --- | --- | --- |
| | | $h^{\mathrm{max}}$ | | $h^{\mathrm{blind}}$ | |
| | | $\mathcal{N}$ | $\mathcal{TN}$ | $\mathcal{N}$ | $\mathcal{TN}$ |
| blocks-4ops | NLL | 1.92 | **1.64** | 1.92 | **1.72** |
| | +clip | 1.89 | | 1.89 | |
| | MSE | 6.02 | **4.94** | 6.02 | **5.02** |
| | +clip | 6.01 | | 6.01 | |
| logistics | NLL | .85 | **.81** | .85 | **.83** |
| | +clip | .85 | | .85 | |
| | MSE | **.65** | .65 | .65 | **.64** |
| | +clip | **.65** | | .65 | |
| satellite | NLL | .69 | **.63** | .69 | **.64** |
| | +clip | .69 | | .69 | |
| | MSE | **1.05** | 1.06 | 1.05 | **1.04** |
| | +clip | **1.05** | | 1.05 | |

Table S9: Test metrics for the HGN models with $l = h^{\mathrm{max}}$ and $l = h^{\mathrm{blind}}$, using the *learn/$h^{\mathrm{FF}}$* configuration. For each experiment configuration, we performed $10^4$ training steps, saving the checkpoints with the best validation NLL loss.

| domain | metric | Weights with best val NLL | | | |
| --- | --- | --- | --- | --- | --- |
| | | $h^{\mathrm{max}}$ | | $h^{\mathrm{blind}}$ | |
| | | $\mathcal{N}$ | $\mathcal{TN}$ | $\mathcal{N}$ | $\mathcal{TN}$ |
| blocks-4ops | NLL | -0.14 | **-0.16** | -0.07 | **-0.15** |
| | +clip | -0.14 | | -0.07 | |
| | MSE | 0.42 | **0.41** | 0.47 | **0.43** |
| | +clip | 0.42 | | 0.47 | |
| logistics | NLL | 0.55 | **0.48** | 0.53 | **0.47** |
| | +clip | 0.55 | | 0.53 | |
| | MSE | 0.48 | **0.46** | 0.47 | **0.45** |
| | +clip | 0.48 | | 0.47 | |
| satellite | NLL | **-0.26** | -0.15 | **-0.33** | -0.13 |
| | +clip | **-0.26** | | **-0.33** | |
| | MSE | 0.60 | **0.59** | **0.58** | 0.64 |
| | +clip | 0.60 | | **0.58** | |

# F Planning Results with Standard Deviations of Node Evaluations

As requested by the reviewer, we created a variant of Table 2 which also shows the standard deviation for the number of evaluated nodes. The std values are obtained by first computing the average number of node evaluations across all instances, and then computing their standard deviation over the seeds, so that the results do not contain the effect of instance-wise deviation.

Table S10: Table 2 with standard deviation of node evaluations.

| domain | $h^{\text{FF}}$ | learn/$h^{\text{FF}}$ | | | fixed/none | | |
| --- | --- | --- | --- | --- | --- | --- | --- |
| | | $\mathcal{N}$ | $\mathcal{N}$+clip | $\mathcal{TN}$ | $\mathcal{N}$ | $\mathcal{N}$+clip | $\mathcal{TN}$ |
| Number of solved instances under $10^4$ evaluations (higher the better) | | | | | | | |
| blocks-4opts | 55 | **97.4±1.14** | **97.4±1.14** | **98±1.87** | **98.6±1.14** | 92±3.32 | 90.2±3.11 |
| logistics | 99 | 93±5.83 | **97.6±1.67** | **97.4±2.70** | 40.4±5.55 | 82.2±5.22 | **90.6±3.91** |
| satellite | 84 | 34.4±7.27 | 55.8±9.26 | **65.2±3.27** | 16.2±4.76 | 48±4.06 | **62.6±8.71** |
| Average node evaluations (smaller the better) | | | | | | | |
| blocks-4opts | 5751 | 656±161 | 656±160 | **560±223** | 640±146 | 1537±473 | **1514±282** |
| logistics | 1031 | 1435±590 | 1077±211 | **953±258** | 6841±648 | 2986±549 | **1830±269** |
| satellite | 3398 | 7227±553 | 5557±899 | **4594±343** | 8792±336 | 6336±332 | **4862±705** |

# G  EXPERIMENTAL RESULTS WITH DIFFERENT TIE-BREAKING STRATEGIES

As previously commented, our $\mathcal{TN}$ approach significantly outperforms the other methods in terms of accuracy (see Table 1). Additionally, as Table 2 shows, $\mathcal{TN}$ also outperforms $\mathcal{N} + clip$ (and $\mathcal{N}$) in terms of planning performance in 5 out of 6 cases, when we consider both coverage and number of node evaluations (this is needed because the coverage metric sometimes saturates to 100%). Nonetheless, although the best $\mathcal{TN}$ model significantly outperforms the $h^{\mathrm{FF}}$ baseline in blocksworld, it obtains similar planning results in logistics and is outperformed by $h^{\mathrm{FF}}$ in satellite. This is a surprising result considering that our approach obtains significantly better MSE than $h^{\mathrm{FF}}$ in all three domains.

We hypothesized that this happens because GBFS is sensitive to changes in node ordering caused by minor float fluctuations in heuristic values. $h^{\mathrm{FF}}$ has no such fluctuations as it always returns an integer. Moreover, $h^{\mathrm{FF}}$ is also a strong baseline for logistics and satellite, as it obtains an MSE lower than 1 in both domains. Therefore, even though heuristics learned by $\mathcal{TN}$ are significantly more accurate *relative to* $h^{\mathrm{FF}}$, the *absolute* heuristic error difference is small, leading to only a minor improvement, which is then masked by degradations from float fluctuations. We believe this is why the improvement in *relative* accuracy by our approach does not necessarily translate into better search performance.

We found a simple yet effective mitigation to this issue that modifies the *tie-breaking* strategy of GBFS, which is known to significantly affect its performance (Asai & Fukunaga, 2017a). According to our hypothesis, when the learned heuristic predicts very similar $h$ values for different nodes, the order between them (which is quite sensitive to minor changes in float values) may not be ideal because the $\mathcal{TN}$ model does not explicitly penalize wrong ordering, unlike learning-to-rank approaches. A simple mitigation is to round $h$ to an integer and then break ties with the $h^{\mathrm{FF}}$ heuristic so that the node expansion order among similar $h$-valued nodes becomes in line with that of the $h^{\mathrm{FF}}$ heuristic, if $h^{\mathrm{FF}}$ provides a good search guidance.

Following (Asai & Fukunaga, 2017a;b), a *sorting strategy* of a best-first search algorithm defines the expansion order of search nodes, and is denoted as a list $[f_1, \ldots, f_n, c_{\mathrm{default}}]$ where each $f_i$ is a function of a state and $c_{\mathrm{default}}$ is a so-called *default tie-breaking criteria*, which is either FIFO, LIFO, Random, or * (unspecified). Nodes are sorted according to $f_1$ and ties are resolved according to the function $f_i$ with smallest $i$ for which nodes have different values. If nodes have the same $f_i$ value for every $i$, ties are then solved with the default strategy $c_{\mathrm{default}}$. For example, the classic A* search algorithm Hart et al. (1968) does not specify the default tie-breaking criteria, and is thus denoted as $[g(s) + h(s), *]$. On the other hand, the Fast Downward Helmert (2006) implementation of A* is denoted as $[g(s) + h(s), h(s), \mathrm{fifo}]$, since it sorts the nodes according to $f(s) = g(s) + h(s)$ first, breaks ties with $h(s)$, and finally implements a FIFO queue, i.e., the first node to be inserted will be expanded first. GBFS with unspecified default tie-breaking is $[h(s), *]$, and Pyperplan implementation of GBFS is $[h(s), \mathrm{fifo}]$.

Table S11 shows the (GBFS) planning results of the NLM model with two different tiebreaking configurations: $[h(s), \mathrm{fifo}]$, as in Table 2, and $[\lfloor h(s) \rfloor, h^{\mathrm{FF}}(s), \mathrm{fifo}]$, where the value of the heuristic $h(s)$ is rounded down to the largest integer and, in the case of a tie, it is resolved by resorting to the FF heuristic.

As expected, the new FF-based tiebreaking strategy significantly improved planning performance whenever $h^{\mathrm{FF}}$ was informative: Performance improved in logistics and satellite, but not in blocksworld, where FF-based tiebreaking performed much worse. Since the $h^{\mathrm{FF}}$ baseline is greatly outperformed by the learned heuristics in blocksworld, it makes sense that relying on $h^{\mathrm{FF}}$ to break ties actually hinders performance.

An important result from these experiments is that learned heuristics and $h^{\mathrm{FF}}$ can complement each other through tie-breaking, instead of learned heuristics just piggy-backing on the latter. This can be observed in Table S11, where $\mathcal{TN}$ with *learn/$h^{\mathrm{FF}}$* and FF-based tiebreaking manages to outperform both the $h^{\mathrm{FF}}$ baseline and non-FF-based tiebreaking (*d*) in logistics and satellite, when both coverage and node evaluations are considered. To the best of our knowledge, despite its simplicity, an approach that employs tie-breaking to combine symbolic and data-driven heuristics into neuro-symbolic, hybrid guidance has not been explored before.

Table S11: Planning results for the NLM models with different tiebreaking strategies, using the *learn/h*$^{\text{FF}}$ and *fixed/none* configurations. *(d)* rows correspond to the **d**efault tiebreaking strategy $[h(s), \text{fifo}]$ of GBFS, whereas *(f)* rows correspond to the the **FF**-based tiebreaking strategy $[\lfloor h(s) \rfloor, h^{\text{FF}}(s), \text{fifo}]$. The *bw*, *lg* and *st* rows represent the blocksworld, logistics and satellite domains, respectively. For each configuration, the best metric among $\mathcal{N}, \mathcal{N}+clip, \mathcal{TN}$ is highlighted in **bold**.

| domain | | $h^{\text{FF}}$ | learn/$h^{\text{FF}}$ | | | fixed/none | | |
|---|---|---|---|---|---|---|---|---|
| | | | $\mathcal{N}$ | $\mathcal{N}$+clip | $\mathcal{TN}$ | $\mathcal{N}$ | $\mathcal{N}$+clip | $\mathcal{TN}$ |
| Number of solved instances under $10^4$ evaluations (higher the better) | | | | | | | | |
| bw | (d) | 55 | **97.4±1.14** | **97.4±1.14** | **98.0±1.87** | **98.6±1.14** | 92.0±3.32 | 90.2±3.11 |
| | (f) | | 95.8±2.86 | 95.0±1.87 | **97.2±2.28** | **98.2±1.92** | 90.6±3.78 | 90.6±2.79 |
| lg | (d) | 99 | 93.0±5.83 | 97.6±1.67 | **97.4±2.70** | 40.4±5.55 | 82.2±5.22 | **90.6±3.91** |
| | (f) | | **99.0±1.00** | **99.4±.89** | 98.6±2.61 | 53.4±7.30 | 94.4±4.56 | **99.2±.84** |
| st | (d) | 84 | 34.4±7.27 | 55.8±9.26 | **65.2±3.27** | 16.2±4.76 | 48.0±4.06 | **62.6±8.71** |
| | (f) | | 58.4±15.61 | 79.2±7.79 | **83.6±1.34** | 47.2±11.78 | 75.0±3.08 | **80.4±5.22** |
| Average node evaluations (smaller the better) | | | | | | | | |
| bw | (d) | 5752 | 656 | 656 | **560** | **640** | 1537 | 1514 |
| | (f) | | 1019 | 1129 | **725** | **626** | 1584 | 1508 |
| lg | (d) | 1032 | 1435 | 1077 | **953** | 6841 | 2986 | **1830** |
| | (f) | | **888** | 890 | 917 | 5591 | 1495 | **977** |
| st | (d) | 3399 | 7227 | 5557 | **4594** | 8792 | 6336 | **4862** |
| | (f) | | 5428 | 3829 | **3158** | 6419 | 4340 | **3687** |

# H  Experimental Results on Large Instances

In this Appendix, we test the generalization abilities of the learned heuristics when evaluated on larger problems than those used during training. We generated a new set of test problems by providing larger parameter values to the instance generators. These new generator parameters are as follows: In blocksworld, we generated problems containing between 11 and 22 blocks. In logistics, we used the following set of parameter values: $airplanes = 3..8$, $cities = 3..8$, $city\ size = 3..6$, $packages = 5..8$, $trucks = 5$. In satellite, we utilized the following: $satellites = 2..7$, $max\ instruments\ per\ satellite = 3..4$, $modes = 4..7$, $targets = [8, 11, 16]$, $observations = 4..7$.

Table S12 shows how the NLM model successfully generalizes to larger problems, with only a small decrease in accuracy when compared to the results for small problems detailed in Table 1. The LR model also manages to generalize to a certain extent, although the MSE obtained for large problems (see Table S13) is moderately higher than the one obtained for small problems (see Table S3). On the other hand, HGN accuracy degrades significantly, i.e., it fails to generalize (see Table S14). For example, in some configurations, the MSE is sometimes larger than 10, i.e., the model mispredicts $h^*$ by more than 10 steps in unit-cost settings, which would render the heuristic function almost unusable.

We ascribe this generalization failure to the neural architecture of the HGN and **not** to the $\mathcal{TN}$ model, since HGN fails even for the $\mathcal{N}$ model (which includes traditional MSE-based training). For example, $\mathcal{N}$ with *fixed/none* in blocksworld obtains an MSE of 31.74. This is natural since generalization mainly depends on the neural architecture employed for learning.

We also present the planning results on these larger instances in Table S15. First, as expected, both the $h^{\mathrm{FF}}$ baseline and learned heuristics resulted in smaller coverage and higher node evaluations overall compared to the results for small problems, since the problem size tends to positively correlate with the difficulty of the instance (though not necessarily).

Second, Table S15 draws a similar conclusion as the one obtained for smaller problems (Table 2): $\mathcal{TN}$ outperforms $\mathcal{N}$ and $\mathcal{N} + clip$ in 4 out of 6 cases. The first failure case is *fixed/none* in Blocksworld, which reflects the lack of accuracy also observed in the smaller instances due to reasons we already discussed in the main paper (*fixed/none* is a problematic configuration as it contains an ad-hoc assumption on $\sigma$ and lacks the normalization by the residual base). The second case is *learn/$h^{\mathrm{FF}}$* in logistics, where $\mathcal{N} + clip$ achieves a minor improvement over $\mathcal{TN}$, of only 0.2 coverage and 20 fewer nodes expanded, which is not a significant difference.

Lastly, if we compare $\mathcal{TN}$ and $h^{\mathrm{FF}}$, the results are almost identical to those obtained for small problems: $\mathcal{TN}$ greatly outperforms $h^{\mathrm{FF}}$ in blocksworld, obtains similar results in logistics (a minor decrease in coverage that is balanced by a moderate decrease in the number of nodes) and is outperformed by $h^{\mathrm{FF}}$ in satellite.

From the results obtained, we conclude that the NLM model is able to successfully generalize to larger problems. Also, since the patterns observed for small problems also arise for larger problems, we believe that employing FF-based tiebreaking will allow our approach to also outperform the $h^{\mathrm{FF}}$ baseline on large problems for all three domains, as shown in Table S11. Finally, upon paper acceptance, we will include the tiebreaking results along with planning results for HGN and LR.

Table S12: Test metrics for NLM on large instances.

| domain | metric | learn/$h^{\mathrm{FF}}$ $\mathcal{N}$ | $\mathcal{TN}$ | learn/none $\mathcal{N}$ | $\mathcal{TN}$ | fixed/$h^{\mathrm{FF}}$ $\mathcal{N}$ | $\mathcal{TN}$ | fixed/none $\mathcal{N}$ | $\mathcal{TN}$ | $h^{\mathrm{FF}}$ |
|---|---|---|---|---|---|---|---|---|---|---|
| blocks-4ops | NLL | .69 | .69 | 1.41 | **1.18** | 1.54 | **.65** | 1.93 | **1.09** | |
| | +clip | **.68** | | 1.30 | | 1.54 | | 1.87 | | |
| | MSE | 1.35 | **1.07** | 3.68 | **3.27** | 1.10 | **1.08** | 2.66 | **2.29** | 22.47 |
| | +clip | 1.35 | | 3.30 | | 1.09 | | 2.41 | | |
| logistics | NLL | 1.98 | **1.08** | 3.07 | **1.01** | 1.44 | **.73** | 6.78 | 2.71 | |
| | +clip | 1.97 | | 1.76 | | 1.44 | | **2.25** | | |
| | MSE | .70 | **.62** | 25.23 | **.60** | .70 | **.64** | 22.05 | **2.46** | 0.89 |
| | +clip | .70 | | 3.92 | | .69 | | 3.93 | | |
| satellite | NLL | 1.08 | **.76** | 1.75 | **.57** | 1.63 | **.56** | 2.33 | **.82** | |
| | +clip | 1.05 | | 1.07 | | 1.58 | | 1.67 | | |
| | MSE | 1.39 | **.64** | 4.29 | **.80** | 1.47 | **1.00** | 4.24 | **1.13** | 1.80 |
| | +clip | 1.28 | | 1.57 | | 1.27 | | 1.61 | | |

Table S13: Test metrics for LR on large instances.

| domain | metric | learn/$h^{\mathrm{FF}}$ $\mathcal{N}$ | $\mathcal{TN}$ | learn/none $\mathcal{N}$ | $\mathcal{TN}$ | fixed/$h^{\mathrm{FF}}$ $\mathcal{N}$ | $\mathcal{TN}$ | fixed/none $\mathcal{N}$ | $\mathcal{TN}$ | $h^{\mathrm{FF}}$ |
|---|---|---|---|---|---|---|---|---|---|---|
| blocks-4ops | NLL | 2.08 | **1.51** | 2.08 | **1.51** | 3.00 | **2.78** | 3.00 | **2.77** | |
| | +clip | 2.06 | | 2.06 | | 2.98 | | 2.98 | | |
| | MSE | 6.49 | 7.22 | 6.52 | **7.23** | 6.94 | 7.81 | 6.93 | 7.75 | 22.47 |
| | +clip | **6.48** | | 6.51 | | **6.86** | | **6.84** | | |
| logistics | NLL | 1.21 | **1.02** | 1.18 | **1.02** | 1.57 | **1.38** | 1.57 | **1.40** | |
| | +clip | 1.21 | | 1.18 | | 1.57 | | 1.57 | | |
| | MSE | 1.06 | **.91** | 1.06 | **.64** | 1.23 | 2.60 | 1.21 | 2.66 | 0.89 |
| | +clip | 1.06 | | 1.06 | | **1.23** | | **1.21** | | |
| satellite | NLL | 1.09 | **.49** | .99 | **.49** | 1.61 | **.74** | 1.58 | **.72** | |
| | +clip | 1.08 | | .99 | | 1.59 | | 1.57 | | |
| | MSE | 1.75 | **.69** | 1.76 | **.70** | 1.40 | 1.43 | 1.27 | 1.38 | 1.80 |
| | +clip | 1.75 | | 1.76 | | **1.31** | | **1.20** | | |

Table S14: Test metrics for HGN on large instances.

| domain | metric | learn/$h^{\mathrm{FF}}$ $\mathcal{N}$ | $\mathcal{TN}$ | learn/none $\mathcal{N}$ | $\mathcal{TN}$ | fixed/$h^{\mathrm{FF}}$ $\mathcal{N}$ | $\mathcal{TN}$ | fixed/none $\mathcal{N}$ | $\mathcal{TN}$ | $h^{\mathrm{FF}}$ |
|---|---|---|---|---|---|---|---|---|---|---|
| blocks-4ops | NLL | **5.29** | 6.44 | 3.03 | 815.63 | 2.03 | **1.52** | 9.20 | **1.06** | |
| | +clip | **5.28** | | **3.00** | | 2.03 | | 9.15 | | |
| | MSE | **1.50** | 1.58 | 3.55 | 22.15 | 3.07 | **2.55** | 31.74 | **1.99** | 22.47 |
| | +clip | **1.50** | | **3.55** | | 3.07 | | 31.54 | | |
| logistics | NLL | 16.66 | 34.48 | 8.06 | 8.32 | 1.56 | **1.17** | 3.35 | **1.90** | |
| | +clip | **14.83** | | **5.11** | | 1.54 | | 2.28 | | |
| | MSE | 1.29 | 1.42 | 5.58 | 37.11 | 1.17 | 1.16 | 8.34 | **2.43** | 0.89 |
| | +clip | **1.23** | | **4.26** | | **1.08** | | 4.05 | | |
| satellite | NLL | 2.82 | **1.52** | 1.05 | .89 | 1.60 | **.44** | 1.58 | **.45** | |
| | +clip | 2.73 | | **.88** | | 1.53 | | 1.52 | | |
| | MSE | 1.18 | **.57** | 1.24 | **.70** | 1.35 | **.77** | 1.25 | **.74** | 1.80 |
| | +clip | .99 | | .93 | | 1.08 | | 1.00 | | |

Table S15: Planning results using NLM on large instances.

| domain | $h^{\mathrm{FF}}$ | learn/$h^{\mathrm{FF}}$ | | | fixed/none | | |
| | | $\mathcal{N}$ | $\mathcal{N}$+clip | $\mathcal{TN}$ | $\mathcal{N}$ | $\mathcal{N}$+clip | $\mathcal{TN}$ |
| --- | --- | --- | --- | --- | --- | --- | --- |
| Number of solved instances under $10^4$ evaluations (higher the better) | | | | | | | |
| blocks-4opts | 54 | 95.4±2.07 | 95.4±2.07 | **97.0±1.22** | **96.8±1.48** | 87.6±4.93 | 88.4±2.51 |
| logistics | 43 | 34.4±8.85 | **40.4±4.51** | **40.2±3.11** | .0±.00 | 19.4±7.89 | **27.4±4.72** |
| satellite | 80 | 21.6±6.54 | 45.0±11.11 | **57.2±2.95** | 4.4±1.67 | 31.6±1.14 | **46.8±8.07** |
| Average node evaluations (smaller the better) | | | | | | | |
| blocks-4opts | 5594 | 871 | 871 | **810** | **1243** | 2031 | 1828 |
| logistics | 4816 | 4870 | **3953** | 3973 | 10000 | 7827 | **6173** |
| satellite | 4711 | 8488 | 6521 | **5622** | 9738 | 7630 | **6547** |

# I  DOMAIN DESCRIPTIONS

In this Appendix, we provide detailed descriptions and PDDL encodings for the three planning domains employed in our experiments: blocksworld-4ops, logistics and satellite.

## I.1  BLOCKSWORLD-4OPS

Blocksworld is one of the oldest domains in the planning literature. It represents a table with a collection of blocks that can be stacked on top of each other. The goal in this domain is to rearrange the blocks to achieve a specific configuration, starting from some initial block arrangement. Blocks can be placed on top of another block or on the table, and every block can never have more than a single block on top of it. The arm/crane used to move the blocks around can only carry a single block at the same time. Listing 1 contains the PDDL description of this domain.

Listing 1: PDDL domain for blocksworld-4ops.

```
(define (domain blocksworld-4ops)
  (:requirements :strips)
(:predicates (clear ?x)
             (on-table ?x)
             (arm-empty)
             (holding ?x)
             (on ?x ?y))

(:action pickup
  :parameters (?ob)
  :precondition (and (clear ?ob) (on-table ?ob) (arm-empty))
  :effect (and (holding ?ob) (not (clear ?ob)) (not (on-table ?ob)
     ) (not (arm-empty))))

(:action putdown
  :parameters (?ob)
  :precondition (holding ?ob)
  :effect (and (clear ?ob) (arm-empty) (on-table ?ob)
               (not (holding ?ob))))

(:action stack
  :parameters (?ob ?underob)
  :precondition (and (clear ?underob) (holding ?ob))
  :effect (and (arm-empty) (clear ?ob) (on ?ob ?underob)
               (not (clear ?underob)) (not (holding ?ob))))

(:action unstack
  :parameters (?ob ?underob)
  :precondition (and (on ?ob ?underob) (clear ?ob) (arm-empty))
  :effect (and (holding ?ob) (clear ?underob)
               (not (on ?ob ?underob)) (not (clear ?ob)) (not (
                  arm-empty)))))
```

## I.2  LOGISTICS

Logistics is another well-known, classical planning domain. It simulates a transportation network where the goal is to move packages from their starting locations to specified destinations. This domain involves several cities, each of them having one or more locations, some of which may be airports. In order to transport the packages, two types of vehicles are available: trucks and airplanes. A truck can move packages between locations within the same city. On the other hand, an airplane can move packages between airports located in different cities. Vehicles can transport an infinite number of

packages at the same time but packages must be loaded and unloaded one at a time. Listing 2 contains the PDDL description of this domain.

Listing 2: PDDL domain for logistics.

```
(define (domain logistics-strips)
  (:requirements :strips)
  (:predicates  (obj ?obj)
                (truck ?truck)
                (location ?loc)
                (airplane ?airplane)
                (city ?city)
                (airport ?airport)
                (at ?obj ?loc)
                (in ?obj1 ?obj2)
                (in-city ?obj ?city))

(:action load-truck
  :parameters (?obj ?truck ?loc)
  :precondition (and (obj ?obj) (truck ?truck) (location ?loc) (at
      ?truck ?loc) (at ?obj ?loc))
  :effect (and (not (at ?obj ?loc)) (in ?obj ?truck)))

(:action load-airplane
  :parameters (?obj ?airplane ?loc)
  :precondition (and (obj ?obj) (airplane ?airplane) (location ?
      loc) (at ?obj ?loc) (at ?airplane ?loc))
  :effect (and (not (at ?obj ?loc)) (in ?obj ?airplane)))

(:action unload-truck
  :parameters (?obj ?truck ?loc)
  :precondition (and (obj ?obj) (truck ?truck) (location ?loc) (at
      ?truck ?loc) (in ?obj ?truck))
  :effect (and (not (in ?obj ?truck)) (at ?obj ?loc)))

(:action unload-airplane
  :parameters (?obj ?airplane ?loc)
  :precondition (and (obj ?obj) (airplane ?airplane) (location ?
      loc) (in ?obj ?airplane) (at ?airplane ?loc))
  :effect (and (not (in ?obj ?airplane)) (at ?obj ?loc)))

(:action drive-truck
  :parameters (?truck ?loc-from ?loc-to ?city)
  :precondition (and (truck ?truck) (location ?loc-from) (location
      ?loc-to) (city ?city) (at ?truck ?loc-from) (in-city ?
      loc-from ?city) (in-city ?loc-to ?city))
  :effect (and (not (at ?truck ?loc-from)) (at ?truck ?loc-to)))

(:action fly-airplane
  :parameters (?airplane ?loc-from ?loc-to)
  :precondition (and (airplane ?airplane) (airport ?loc-from) (
      airport ?loc-to) (at ?airplane ?loc-from))
  :effect (and (not (at ?airplane ?loc-from)) (at ?airplane ?
      loc-to)))
)
```

## I.3    SATELLITE

Satellite is another domain that is widely employed in the planning literature. It simulates the operation of one or more satellites in space. The goal involves using the satellites to collect images in specific modes and orientations, requiring careful management of resources like power and instrument capabilities. Actions in this domain include orienting the satellite, switching instruments on or off, calibrating instruments for specific tasks, and taking images. Key challenges include managing the limited power available to the satellite, ensuring instruments are correctly calibrated for their tasks, and aligning the satellite to point in the correct direction for each task. Listing 3 contains the PDDL description of this domain.

Listing 3: PDDL domain for satellite.

```
(define (domain satellite)
  (:requirements :strips :typing)
  (:types satellite direction instrument mode)
  (:predicates
        (on_board ?i - instrument ?s - satellite)
        (supports ?i - instrument ?m - mode)
        (pointing ?s - satellite ?d - direction)
        (power_avail ?s - satellite)
        (power_on ?i - instrument)
        (calibrated ?i - instrument)
        (have_image ?d - direction ?m - mode)
        (calibration_target ?i - instrument ?d - direction))

  (:action turn_to
   :parameters (?s - satellite ?d_new - direction ?d_prev -
      direction)
   :precondition (and (pointing ?s ?d_prev))
   :effect (and  (pointing ?s ?d_new)
                 (not (pointing ?s ?d_prev))))

  (:action switch_on
   :parameters (?i - instrument ?s - satellite)
   :precondition (and (on_board ?i ?s)
                      (power_avail ?s))
   :effect (and (power_on ?i)
                (not (calibrated ?i))
                (not (power_avail ?s))))

  (:action switch_off
   :parameters (?i - instrument ?s - satellite)
   :precondition (and (on_board ?i ?s)
                      (power_on ?i))
   :effect (and (not (power_on ?i))
                (power_avail ?s)))

  (:action calibrate
   :parameters (?s - satellite ?i - instrument ?d - direction)
   :precondition (and (on_board ?i ?s)
                      (calibration_target ?i ?d)
                      (pointing ?s ?d)
                      (power_on ?i))
   :effect (calibrated ?i))

  (:action take_image
   :parameters (?s - satellite ?d - direction ?i - instrument ?m -
      mode)
   :precondition (and (calibrated ?i)
                      (on_board ?i ?s)
                      (supports ?i ?m)
                      (power_on ?i)
                      (pointing ?s ?d))
   :effect (have_image ?d ?m)))
```

