# OpenReview forum: "On Using Admissible Bounds for Learning Forward Search Heuristics"
_ICLR.cc/2024/Conference — Submitted to ICLR 2024_

### Official Review · Reviewer_j4wF · 2023-10-15

**Soundness:** 2 fair
**Presentation:** 3 good
**Contribution:** 2 fair
**Rating:** 5
**Confidence:** 3

**Summary:**

This paper proposes to learn planning heuristics for forward search algorithms. The authors propose using truncated Gaussians to model the distribution of the learned heuristics. This modeling change results in a different loss function from the standard MSE loss. Empirical evaluations are provided in three classical planning domains.

**Strengths:**

1. I found the exposition interesting: connecting the MSE loss with the assumption that the empirical distribution is a Gaussian distribution with a fixed variance motivates the proposed method well.

2. Employing existing knowledge to obtain lower bounds to further constrain the estimated distribution is a simple and elegant method to make the predictions more informative. The empirical evidence shows such an additional constraint improves the search efficiency.

**Weaknesses:**

1. The choice of using Greedy Best-first Search (GBFS) needs more justification as it is not guaranteed to find optimal solutions. Can the authors explain why they did not choose to experiment with A*?

2. Overall the empirical results are quite weak. While the negative log-likelihood (NLL) and MSE losses improve, the downstream search performance, i.e., the average number of node evaluations, compared to baselines (either the non-truncated Gaussian or using $h^{FF}$ directly) is not convincing.

3. A popular direction for evaluating the effectiveness of a learned heuristic is to test how well it generalizes to larger instances of a class of problems. The current evaluations focus only on problems generated with the same set of parameters. Adding this kind of generalization experiment would demonstrate the practical value of the proposed approach better.

4. The three classical planning problems should have sufficiently detailed descriptions.

**Questions:**

Please see the above section for my question regarding the choice of the search algorithm.

Additionally, can the authors add the standard deviation numbers for the average number of node evaluations to Table 2?

---

> ### Author Response · Authors · 2023-11-19
>
> Thank you very much for your valuable feedback and questions.
> We are pleased to know you believe our method is well-motivated,
> _"simple"_ and _"elegant"_.
> We now answer the questions and concerns you have raised.
>
> > The choice of using Greedy Best-first Search (GBFS) needs more justification as it is not guaranteed to find optimal solutions. Can the authors explain why they did not choose to experiment with A*?
>
> The reason for choosing GBFS over A\* is that the learned heuristic $\hat{h}$
> is not guaranteed to be **admissible**, regardless of the ML method employed
> (NLM, LR or HGN) and how the heuristic is modelled ($\mathcal{N}$ vs $\mathcal{TN}$).
> An admissible heuristic is one that never exceeds the optimal plan cost
> $h^*$ for any state: $\forall s \in S, \hat{h}(s) \leq h^*(s)$.
> Unlike symbolic algorithms, machine learning methods (e.g., neural networks) do not provide such guarantees in general.
> It is important to note that this issue is not particular to our approach since, to the best of our knowledge, **all heuristics learned by ML methods lack admissibility**.
>
> The A\* algorithm is guaranteed to return an optimal plan *only when* the heuristic used to guide the search is admissible (Russell and Norvig, 2010, pp. 94-95). Since $\hat{h}$ is inadmissible, we cannot guarantee solution optimality regardless of the search algorithm employed (A\* or GBFS).
> Therefore, we disregard plan length altogether and evaluate search performance as the number of node evaluations needed to find a solution, i.e.,
> focus on satisficing planning.
>
> In satisficing planning, A\* is known to often be slower than WA\* and GBFS (corresponding to WA\* with $W\rightarrow\infty$) because it needs to explore all the nodes below the best current $f=g+h$ value, while WA\* and GBFS do not.
> This is the main reason why our work and many others in the heuristic learning literature (Ferber et al., 2022; Gehring et al., 2022; Yoon et al., 2008) employ GBFS instead of A*. **A* is not an appropriate baseline for the satisficing setting.**
>
> > Overall the empirical results are quite weak. While the negative log-likelihood (NLL) and MSE losses improve, the downstream search performance, i.e., the average number of node evaluations, compared to baselines (either the non-truncated Gaussian or using directly) is not convincing.
>
> We address this concern in the common answer to all reviewers (additional tiebreaking experiments).
>
> > A popular direction for evaluating the effectiveness of a learned heuristic is to test how well it generalizes to larger instances of a class of problems. The current evaluations focus only on problems generated with the same set of parameters. Adding this kind of generalization experiment would demonstrate the practical value of the proposed approach better.
>
> We have performed new experiments where we test the generalization abilities of the learned heuristics on larger problems. An overall analysis of the results obtained can be found in the common answer to all reviewers.
>
> > The three classical planning problems should have sufficiently detailed descriptions.
>
> We have added detailed descriptions for the three domains (blocksworld, logistics and satellite) used in our paper, along with their PDDL encodings.
> This information can now be found in Appendix I.
>
> > Can the authors add the standard deviation numbers for the average number of node evaluations to Table 2?
>
> We have added the std of node evaluations, as requested.
> This new information can be found in Appendix F.
>
> We hope to have successfully addressed all your questions and concerns with our answers and paper additions.

---

> ### Comment · Reviewer_j4wF · 2023-11-22
> **Thanks for the response**
>
> I want to thank the authors for their detailed responses to my questions. I am glad to see the new tie-break strategy helps improve the results. However, I still have concerns with the empirical results. The generalization results in the Appendix only concern the NLL and MSE without planning results. Combined with the fact the three planning tasks are quite artificial, I think the overall evaluation could be strengthened. I will raise my score to 5.

---

> > ### Author Response · Authors · 2023-11-22
> >
> > Thank you for an additional response as well as revision to your score. We now answer your new concerns.
> >
> > ## Generalization experiments
> > As requested, we updated the paper with new results and analysis (table S15, see page 28, Appendix H) that shows the planning results also generalizes to larger instances.
> >
> > We did not include it in the first reply due to the time constraints: we were only able to run experiments for the NLM model and without FF-based tiebreaking (new enhancements added in the first reply). We are also running the other configurations (HGN, LR, with/without tiebrekaing, on large instances).
> >
> > The new results support our claim that $\mathcal{TN}$+NLM is able to generalize to larger problems.
> > The (relative) results on large problems are almost identical to those obtained for small problems: $\mathcal{TN}$ outperforms $\mathcal{N}$ and $\mathcal{N}+clip$ in 4 out of 6 cases, with one case obtaining similar performance to $\mathcal{N}+clip$ and being outperformed by $\mathcal{N}$ in blocksworld with fixed/none (this also occurred for small problems, since this is a problematic configuration due to the fixed $\sigma$ and no residual learning). When comparing to the FF baseline, the results are also the same: $\mathcal{TN}$ greatly outperforms FF in blocksworld, obtains similar results in logistics and is outperformed in satellite.
> >
> > Given that the results for small and large problems are very similar, we believe that, just as it occurred for small problems, employing FF-based tiebreaking will allow out approach to outperform the FF baseline on large problems for all three domains. We will add the missing experiments (tiebreaking and results for HGN and LR) as soon as we have time to run these experiments.
> >
> > ## PDDL domains are NOT toy tasks
> > The Satellite domain used in our evaluaiton, which first appeared in International Planning Competition 2002 (https://ipc02.icaps-conference.org/domains.html), is a domain inspired from NASA's real observatory missions including space probes like Deep Space One, planetary rovers like Mars Sojourner, space-based observatories like the Hubble Space Telescope, airborne observatories like KAO and SOFIA (Smith et. al., 2000).
> >
> > It is a standard practice to evaluate planning performance on PDDL domains. Many of these domains are directly compiled from actual industrial applications (e.g., PARCPrinter domain optimizing XEROX Parc automated printers, BAMS cybersecurity domains, etc. https://ipc08.icaps-conference.org/deterministic/Domains.html) The International Planning Competitions (IPCs) (https://www.icaps-conference.org/competitions/) which are held bi-annualy, use PDDL domains like the ones in our work for comparing the performance of state-of-the-art planners.
> >
> > We hope to have successfully addressed your new concerns.
> >
> > Smith, David E., Jeremy Frank, and Ari K. Jónsson. "Bridging the gap between planning and scheduling." The Knowledge Engineering Review 15.1 (2000): 47-83.

---

### Official Review · Reviewer_hjVN · 2023-10-17

**Soundness:** 2 fair
**Presentation:** 3 good
**Contribution:** 1 poor
**Rating:** 3
**Confidence:** 2

**Summary:**

This paper studies heuristic learning by utilizing the information provided from admissible heuristics as informative bounds. To this end, instead of the traditional approach of minimizing mean square errors, the authors propose to model the learned heuristic as a truncated gaussian and subsequently minimize the loss function resulted from such a statistical/distributional assumption. As such, the authors claim to provide some theoretical understanding which is often lacking in the past work. Experiments are conducted and they show that the proposed method does contain certain merits.

**Strengths:**

The presentation is very clear and overall, the paper is easy to follow and digest. The proposed method also makes intuitive sense. Given that there are additional information available, it makes sense to utilize them in the modeling instead of going with the traditional gaussian case. The experiments also, to certain extent, verify the founding.

**Weaknesses:**

The contributions of this paper may seem weak and limited. For example, the connection among MLE, Gaussian, MSE is well-known and Section 3 seems to be elementary. If admissible bounds are available, it seems straightforward/natural to refine the distributional assumption. Overall, I think this paper contains a good practical study but it doesn't seem to be innovative enough to justify the acceptance.

**Questions:**

Could the authors provide some discussions on how certain assumptions in the current manuscript would change the result/model? For example, "In this paper, we assume unit-cost: ∀a ∈ A; COST(a) = 1;" "In this work, we focus on the scenario where an admissible heuristic is provided along with the optimal solution cost h∗ for each state, leaving other settings for future work."

---

> ### Author Response · Authors · 2023-11-19
> **General reply**
>
> Thank you very much for your valuable feedback and comments.
> We are glad you found our paper _"easy to follow"_ and believe
> our approach _"makes intuitive sense"_.
> We now answer the questions and concerns you have raised.
>
> > the connection among MLE, Gaussian, MSE is well-known and Section 3 seems to be elementary.
>
> That is correct, since the aim of Section 3 is providing an extended background rather than explaining our contributions. This background is intended to serve as context for researchers who may not be familiar with this statistical knowledge (e.g., from the AI planning community).
> Section 4 builds upon this background to present what is the **main contribution** of our paper: using admissible heuristics as lower bounds of a $\mathcal{TN}$ distribution modelling the learned heuristic.

---

> ### Author Response · Authors · 2023-11-19
> **Discussion about novelty**
>
> > it seems straightforward/natural ...
>
> Using admissible heuristics as lower bounds of a $\mathcal{TN}$ distribution may appear trivial in the hindsight, but it is not straightforward nor obvious.
>
> While there exist several works that leverage $\mathcal{TN}$ for machine learning,
> their usage is most often discussed in the context of safety-aware planning.
> For example,
> (Murray et. al, 2023) uses a $\mathcal{TN}$ to model a Simple Temporal Network with Uncertainty (STNU) which can model a distribution of time within a specific start time / deadline.
> (Eisen et al, 2019) uses a $\mathcal{TN}$ to optimize wireless device allocations, where the truncation encodes the minimum (0) and maximum signal power output.
> In robotics, it is common to use $\mathcal{TN}$ for exploration (Chen et al, 2018) or
> to limit the measurement uncertainty of acceleration, velocity, positions (Kamran et al, 2018).
> In all cases, upper/lower bounds are treated as a **static** and **arbitrary** constraint imposed by the environment or by a human operator.
>
> In contrast,
> admissible heuristics used as lower bounds in our work
> are **formal bounds automatically derived** by symbolic algorithms
> and are **dynamic**, as their value depend on each particular state
> explored by a search algorithm (e.g., A\*).
> For example, Landmark-cut heuristic (Helmert, Domshlak, 2009) is computed by
> deriving a so-called landmark graph, then
> iteratively reduces the edge costs in this graph on the edges that constitute a cut of the graph.
>
> To our knowledge, **our work is first to show that such symbolic, dynamic, formally derived bounds can be combined with a $\mathcal{TN}$ in order to aid training.**
>
> In applications of machine learning to Operations Research problems (Vehicle routing problem, TSP),
> most of the existing work tries to learn to solve the problems from the scratch without the help of heuristics (e.g., Nazari et al, 2018).
> Although some work (Liang et al, 2021) uses the optimal solution obtained by traditional methods (e.g. Concorde) for training and
> combines it with existing admissible heuristics (LKH heuristic) during testing,
> it does not use the heuristic for training nor as the lower-bound of a truncated Gaussian distribution.
>
> In the context of heuristic learning for planning,
> the only role proposed so far for symbolically-derived heuristics
> is as a training target (which, as discussed in Section 4, is not a good approach)
> or as a basis for residual learning (Yoon et al, 2009).
> The situation is similar in the Reinforcement Learning literature.
> While there exist works that help value/Q-function learning through reward shaping,
> which is theoretically equivalent to residual learning (Ng, Harada, Russel, 1999),
> and its extension (Ching-An et. al., 2021, HuRL) which utilizes domain-specific hand-crafted heuristics,
> none has leveraged heuristics as lower/upper bounds for improving learning so far.
> Ching-An et. al. also discussed the relation of pessimistic and admissible heuristics
> as desirable properties of RL and planning heuristics,
> but their method does not explicitly use the upper/lower bound property for training.
>
> Finally, the novelty of our work is also supported by the comments of Reviewers LPcd and 795y, the latter claiming that our paper _successfully challenges the accepted wisdom of using MSE when learning heuristics_.
>
> * Helmert, Malte, and Carmel Domshlak. "Landmarks, critical paths and abstractions: what's the difference anyway?." Proceedings of the International Conference on Automated Planning and Scheduling. Vol. 19. 2009.
> * Murray, Andrew, et al. "A column generation approach to correlated simple temporal networks." Proceedings of the International Conference on Automated Planning and Scheduling. Vol. 33. No. 1. 2023.
> * Eisen, Mark, et al. "Learning optimal resource allocations in wireless systems." IEEE Transactions on Signal Processing 67.10 (2019): 2775-2790.
> * Chen, Tao, Saurabh Gupta, and Abhinav Gupta. "Learning Exploration Policies for Navigation." International Conference on Learning Representations. 2018.
> * Kamran, Danial, et al. "Minimizing safety interference for safe and comfortable automated driving with distributional reinforcement learning." 2021 IEEE/RSJ International Conference on Intelligent Robots and Systems (IROS). IEEE, 2021.
> * Nazari, Mohammadreza, et al. "Reinforcement learning for solving the vehicle routing problem." Advances in neural information processing systems 31 (2018).
> * Xin, Liang, et al. "NeuroLKH: Combining deep learning model with Lin-Kernighan-Helsgaun heuristic for solving the traveling salesman problem." Advances in Neural Information Processing Systems 34 (2021).
> * Ng, Andrew Y., Daishi Harada, and Stuart Russell. "Policy invariance under reward transformations: Theory and application to reward shaping." Icml. Vol. 99. 1999.
> * Cheng, Ching-An, Andrey Kolobov, and Adith Swaminathan. "Heuristic-guided reinforcement learning." Advances in Neural Information Processing Systems 34 (2021).

---

> ### Author Response · Authors · 2023-11-19
> **Effects of our assumptions on our approach and obtained results**
>
> > "In this paper, we assume unit-cost: ∀a ∈ A; COST(a) = 1;"
>
> This assumption does not affect our work in any way,
> as our method is completely agnostic to it.
> We assumed unitary costs because most _PDDL_ domains only contain unitary-cost actions.
> Our training method only requires training data (i.e., $h^*$, $l$ and heuristic values for residual learning) to be represented as real numbers,
> and is indifferent to whether it originates from unitary or variable cost domains.
>
> > "we focus on the scenario where an admissible heuristic is provided along with the optimal solution cost $h^∗$"
>
> In this work, we have focused on the case where both $l$ and $h^*$ are available, since this is the most common scenario in the cost-to-go learning literature (i.e., having a dataset with optimal costs and being able to compute domain-independent, admissible heuristics).
> This case also includes the scenario where $l$ is not available but only $h^*$, since we can always use $l=h^{blind}$ or $l=0$.
>
> Other scenarios have been left out as future work (see Section 7, Paragraph 3).
> One such scenario is when the upper bound $u$ is available in addition to $lb$. Our approach is perfectly compatible with this scenario and no modifications would be needed. Regarding results, we would expect an increase in performance due to the tighter interval for $\mathcal{TN}$ provided by both $l$ and $u$ when compared to only $l$.
> Another interesting scenario is when $h^*$ is not available. We are working on tackling this setting by using _Variational Inference_ and treating $h^*$ as a hidden variable. This would require some modifications to our method, such as the number of ML models trained and the loss function to optimize (now corresponding to the _ELBO_ instead of the _NLL_).

---

### Official Review · Reviewer_795y · 2023-10-30

**Soundness:** 4 excellent
**Presentation:** 4 excellent
**Contribution:** 4 excellent
**Rating:** 8
**Confidence:** 3

**Summary:**

The paper proposes a new loss function for learning admissible heuristics in AI
search. The authors argue that the widely-used MSE does not accurately model
what we intend to optimize, describe their new loss function, and evaluate it
empirically.

**Strengths:**

The paper is well-written, the presented argument is convincing, and the
empirical results further support it. This is a great paper that successfully
challenges the accepted wisdom of using MSE when learning heuristics.

**Weaknesses:**

I have no concerns or questions regarding the work.

**Questions:**

I have no concerns or questions regarding the work.

---

> ### Author Response · Authors · 2023-11-19
>
> Thank you very much for your encouraging words and valuable feedback.
> We are deeply pleased to know you have enjoyed our paper and found it _"well-written"_, _"convincing"_ and note that _"the empirical results further support it"_.
> We are delighted with your statement that _"this is a great paper that successfully challenges the accepted wisdom of using MSE when learning heuristics"_.

---

### Official Review · Reviewer_LPcd · 2023-10-31

**Soundness:** 3 good
**Presentation:** 3 good
**Contribution:** 3 good
**Rating:** 6
**Confidence:** 4

**Summary:**

The paper presents a new approach for learning heuristic for forward search (specifically, for greedy best-first search) that can make use of admissible heuristics. The proposed approach is based on modelling the learned heuristic using truncated Gaussian and use the admissible estimates as lower bound for the distribution. The best approach that consists of using truncated Gaussian, learned $\sigma$, and residual learning (based on the popular h^FF heuristic) significantly outperform other learning-based configurations in terms of accuracy and number of solved instances.

**Strengths:**

Strengths:
- Novel approach for learning heuristics that makes principled use of admissible estimates.
- The approach is compatible with residual heuristic learning and is agnostic of the neural architecture.
- Experiments show increased accuracy and a larger number of planning instances solved under 10^4 evaluations.

**Weaknesses:**

Weaknesses:
- Missing state-of-the-art recent baseline: [Chrestien et al., 2022] is an alternative approach that also argues against using MSE and proposes an alternative approach. This baseline should be compared to the proposed approach for learning heuristics.
- In the experiments, it looks that the standard h^FF baseline outperforms the proposed approach in terms of problem solved in two out of the three domains.
- There is no analysis of performance vs. problem size. It would be very useful to see if the patterns depend on problem size.

Minor point: in Section 3, the description of learning heuristic ignores the goal, the learning of heuristics is conditioned on the goal state since the same state will have different estimate conditioned on different states.

**Questions:**

I would appreciate the authors' response to the weaknesses listed above. In particular, did you compare to [Chrestien et al., 2022] or other state-of-the-art approaches for learning heuristics?

---

> ### Author Response · Authors · 2023-11-19
>
> Thank you very much for your valuable feedback and insightful comments.
> We are pleased to know you found our method _"novel"_ and _"principled"_.
> We address your comments and concerns below.
>
> > a new approach for learning heuristic for forward search (specifically, for greedy best-first search)
>
> Just a small clarification. The heuristic learned with our method is not tailored to any specific search algorithm (e.g., GBFS) and can in principle be applied to other algorithms such as A\*. We use GBFS in our experiments because the heuristic learned with our approach (or any other ML method) is never guaranteed to be admissible and, thus, regardless of the search algorithm plan optimality is not guaranteed. Therefore, we disregard plan length altogether and focus on satisficing planning, i.e., finding a solution with as few node evaluations as possible. In satisficing search, A\* is slower than WA\* and GBFS (which corresponds to WA* with $W\rightarrow\infty$) because A\* must explore all nodes below the current best $f=g+h$ value.
>
> > Missing comparison with (Chrestien et al., 2022)
>
> Thank you for highlighting this interesting work. We would like to note that (Chrestien et al., 2022) follows the learning-to-rank approach initiated by (Garrett et al., 2016) which learns the relative quality between states, with an additional loss for enforcing heuristic monotonicity. We believe the method proposed in (Chrestien et al., 2022) corresponds to a learning-to-rank approach because the proposed L\* loss function tries to rank the states expanded by A\* before those which are not (as well as ranking them according to monotonicity), modeling the learning task as a binary classification rather than a regression as in our work.
>
> We want to emphasize that **learning-to-rank methods deviate significantly from the cost-to-go learning literature** (which our work belongs to), both empirically and theoretically, as they pursue a different goal (correct node ordering vs correct cost-to-go estimation) and, thus, are significantly outside the scope of our work.
> Our paper aims at addresssing the lack of formal understanding in the cost-to-go learning literature and, in section 3, paragraph 8, we explicitly state that our goal is to learn the cost-to-go $p^*(x|s)$.
>
> Learning-to-rank methods directly optimize search performance by learning the node ordering. On the other hand, cost-to-go approaches learn to estimate the distance from some particular state to the goal, a unique capability missing from ranking-based methods. Being able to estimate the cost-to-go is useful in many practical applications, e.g., knowing the estimated travel time to destination in a mobile navigation app (e.g., Google Maps), as opposed to just
> knowing which direction to go at every intersection (equivalent to ranking the directions).
>
> Applying our theoretical intuition from $\mathcal{TN}$ to the learning-to-rank approach
> is an interesting direction for future work.
>
> > In the experiments, it looks that the standard $h^{FF}$ baseline outperforms the proposed approach in terms of problem solved in two out of the three domains.
>
> We address this concern in the common answer to all reviewers (new tiebreaking experiments).
>
> > There is no analysis of performance vs. problem size. It would be very useful to see if the patterns depend on problem size.
>
> We have performed new experiments where we test the generalization abilities of the learned heuristics on larger problems. An overall analysis of the results obtained can be found in the common answer to all reviewers.
>
> > in Section 3, the description of learning heuristic ignores the goal
>
> Thank you for bringing this oversight to our attention. It has now been corrected in the revision, by stating that heuristics depend on both states and goals.

---

> > ### Comment · Reviewer_LPcd · 2023-11-23
> > **Thank you**
> >
> > Thank your for your response.
> >
> > - My comment on GBFS is indeed due to the lack of admissibility of the learned heuristics.
> >
> > - Regarding (Chrestien et al., 2022): it is true that the setting of the two papers is different, however ultimately the goal of learning heuristics is to use them for planning. Given that the proposed approach is not learning admissible heuristic and is evaluated primarily in GBFS based on coverage, it is not clear to me why it should not be compared with (Chrestien et al., 2022) or other approaches that are focused on learning heuristics that achieve better coverage (for example, (Chrestien et al., 2022) did compare with L2 loss). If there is an inherent benefit for learning cost-to-go (in terms of planning performance) it should be demonstrated.
> >
> > - Results on large instances are encouraging, although it seems that h^FF still outperforms the proposed approach in terms of coverage in two of the domains. I think the new results about tie-breaking are interesting, but are not entirely convincing. It seems that you are creating many ties and then use h^FF to break ties and in the end achieve similar coverage to h^FF. If we consider a simple baseline that just predict a constant value and break ties using h^FF it will be able to get similar coverage to h^FF - therefore it is difficult to really observe the benefit of the proposed approach in terms of planning performance based on this experiment.
> >
> > Overall, I remain positive about the paper.

---

> > > ### Author Response · Authors · 2023-11-23
> > >
> > > Thank you for your new response. We answer your additional comments below.
> > >
> > > ## Re: comparing to (Chrestien et al., 2022)
> > >
> > > The main contribution of our paper is clear: modelling heuristics as $\mathcal{TN}$ (with admissible heuristics as lower bounds) instead of $\mathcal{N}$ results in better heuristics.
> > > Therefore, **all of our experiments are performed to support this claim.**
> > >
> > > With this in mind, **it is unclear what comparing with (Chrestien et al., 2022) would hope to achieve**. Comparing with this learning-to-rank method would not result useful
> > > for proving (or disproving) our claim that "$\mathcal{TN}$ is better than $\mathcal{N}$".
> > > Also, as explained in our previous answer, **cost-to-go and learning-to-rank methods pursue different goals** (accurate heuristic estimation vs correct node
> > > ordering in planning algorithms). When we use $\mathcal{TN}$ instead of $\mathcal{N}$, we change how the heuristic is modelled, but **our goal is still the same**: to accurately predict the optimal cost-to-go $h^*$. However, using the L* loss proposed in (Chrestien et al., 2022) instead of MSE, **completely changes the goal of the model**: it now aims to rank nodes correctly, even if this results in worse heuristic accuracy. We believe this is why (Chrestien et al., 2022) does not show the accuracy (MSE) of the heuristics learned
> > > with L* (which would correspond to Table 1 in our paper), since accuracy is no longer a concern but only correct ordering. Conversely, **accuracy is the main concern of our approach**, just as it happens to be for the cost-to-go learning literature. Regarding your question
> > >
> > > > If there is an inherent benefit for learning cost-to-go (in terms of planning performance) it should be demonstrated.
> > >
> > > Ranking-based approaches (like (Chrestien et al., 2022)) directly optimize planning performance at the expense of being able to accurately predict $h^*$, which is the main goal
> > > of cost-to-go learning approaches like ours. As explained in our previous answer, predicting $h^*$ has many applications, e.g., knowing the estimated travel time in a mobile navigation app. **Comparing the planning performance of cost-to-go and ranking-based approaches is unfair, since cost-based approaches aim at maximizing accuracy whereas ranking-based approaches aim at minimizing incorrect orderings**. It would be equally unfair to compare the accuracy of cost-based vs ranking-based approaches, which is why we believe the authors of (Chrestien et al., 2022) opted for not showing the accuracy of their novel L* loss in their paper.
> > >
> > > Finally, it is misleading to talk about our $\mathcal{TN}$ approach versus the L* loss proposed in (Chrestien et al., 2022), as we believe **both approaches are compatible and complementary**, in the same way $\mathcal{TN}$ can be employed alongside residual learning. In future work, we will study how to leverage admissible heuristics in a statistically consistent way for the learning-to-rank setting.
> > >
> > > ## Re: planning results
> > >
> > > > Results on large instances are encouraging, although it seems that h^FF still outperforms the proposed approach in terms of coverage in two of the domains
> > >
> > > We agree that $h^{FF}$ still obtains better coverage than our approach in two domains for large problems, but this is because the **results in Table S15 do not include FF-based tiebreaking**, since
> > > we did not have time to run the tiebreaking experiments. Just as it occurred for small problems (see Table S11), we believe FF-based tiebreaking will allow our approach to
> > > outperform $h^{FF}$ in all three domains.
> > >
> > > Regarding your comment about FF-based tiebreaking
> > >
> > > > It seems that you are creating many ties and then use $h^{FF}$ to break ties and in the end achieve similar coverage to $h^{FF}$
> > >
> > > we want to emphasize that, as explained in the last paragraph of Appendix G,
> > > **$\mathcal{TN}$+FF-based tiebreaking does not piggy-back on $h^{FF}$ but, instead, outperforms this baseline**.
> > > Firstly, **in blocksworld, this model obtains much better coverage (97.2 vs 55) and number of nodes (725 vs 5752) than the $h^{FF}$ baseline**,
> > > showing how FF-based tiebreaking manages to outperform $h^{FF}$.
> > > Regarding logistics and satellite, it is true that FF-based tiebreaking obtains similar coverage than $h^{FF}$ in these domains, but it manages to outperform
> > > $h^{FF}$ in the number of node evaluations (917 vs 1032 in logistics and 3158 vs 3399 in satellite).
> > > Considering both metrics is especially important in logistics, where coverage does saturate.
> > > In conclusion, when both coverage and number of nodes are considered, **FF-based tiebreaking outperforms the $h^{FF}$ baseline in all three domains**,
> > > showing the superiority of $\mathcal{TN}$+FF-based tiebreaking instead of simply predicting $h=h^{FF}$.
> > >
> > > We hope our answers have successfully addressed your concerns.

---

### Official Review · Reviewer_63nm · 2023-11-05

**Soundness:** 3 good
**Presentation:** 3 good
**Contribution:** 2 fair
**Rating:** 5
**Confidence:** 3

**Summary:**

This paper discusses the use of modern machine learning techniques to learn heuristic functions for forward search algorithms. It highlights the lack of theoretical understanding regarding what these heuristics should learn, how to train them, and why they are trained in the first place, leading to a variety of training targets and loss functions in the literature. The authors argue that learning from admissible heuristics using mean square errors (MSE) as the loss function is not the correct approach because it results in a noisy, inadmissible heuristic. Instead, they propose modeling the learned heuristic as a truncated Gaussian, with admissible heuristics used as lower bounds. This approach results in a different loss function than MSE, leading to faster training convergence and better heuristics, with a 40 percent lower average MSE.

**Strengths:**

1. The paper is enjoyable to read and fairly well organized by raising three core questions (what, how and why). The problem of choice of training target and loss function is also well-motivated.

2. The experiments are relatively comprehensive and the results are pleasing.

**Weaknesses:**

1. In my view, one of the main contributions of this paper is to use the NLL as their training loss instead of MSE, and NLL adds the prediction of $\sigma$ (the variance) where MSE uses the fixed $\sigma$. However, in my opinion, this technique will improve the performance very trivially since the model will predict better with more parameters.

2. The authors explain the reason for using Gaussian distribution by giving the principle of maximum entropy, but the reason for using Truncated Gaussian seems missing and
unconvincing.

**Questions:**

In section 3, the authors claim that the importance of using NLL loss function instead of MSE. However, from the experiment results, the authors can only claim that $\mathcal{T} \mathcal{N}$ obtains around 40 percent lower MSE than $\mathcal{N}$ +clip on (geometric) average, so does this truncated Gaussian technique play the major role of the outstanding performance of the experiment instead of the choice of loss function? And what is the statistical intuition behind it?

---

> ### Author Response · Authors · 2023-11-19
>
> Thank you very much for your valuable feedback and comments.
> We are glad you enjoyed reading our work.
> We now answer the questions and concerns you have raised.
>
> ### Re: Question 1
>
> Firstly, we would like to clarify a point of a slight confusion.
> Your comment mentions as follows:
>
> > In my view, ... to use the NLL as their training loss **instead of** MSE, ...
>
> > In section 3 ... the importance of using NLL loss function **instead of** MSE.
>
> Let us clarify that "use NLL **instead of** MSE" involves a slight misnomer,
> as the MSE is indeed a NLL for $\mathcal{N}(\mu, \sigma=1/\sqrt{2})$.
> In our paper, we wrote:
>
> > [Main paper, Section 2, Paragraph 5] We emphasize that the choice of the distribution determines the loss.
>
> This means that there is a 1-to-1 correspondence between the distribution used and the loss to minimize, i.e.,
> they are **_two sides of the same coin_**.
> Therefore, as to the question
>
> > ... does this truncated Gaussian technique play the major role ... instead of the choice of loss function?
>
> We answer that _the choice of our loss function is a direct consequence of choosing the truncated Gaussian distribution_,
> therefore it is not possible to choose them separately.
> For example, using a truncated Gaussian while minimizing a standard MSE loss
> is impossible/contradicting
> (minimizing MSE implies using $\mathcal{N}(\mu, \sigma=1/\sqrt{2})$).
>
> Likewise, switching modelling assumptions during training and evaluation
> (e.g., train the model with $\mathcal{N}(\mu, \sigma)$, then reuse the network for $\mathcal{TN}(\mu, \sigma, l, \infty)$) is highly ad-hoc and should not be done.
>
>
> In addition, regarding this comment:
>
> > the authors can only claim that $\mathcal{TN}$ obtains around 40 percent lower MSE than $\mathcal{N}+clip$ on (geometric) average, ...
>
> we would like to mention that predictions from the $\mathcal{N}+clip$ model are never worse than those of $\mathcal{N}$, since $\mathcal{N}+clip$ prevents incorrect
> heuristic predictions that are smaller than the admissible heuristic.
> Therefore, we can not only claim that $\mathcal{TN}$ is better than $\mathcal{N}+clip$, but also claim that it is better than $\mathcal{N}$ too.
>
> ### Re: Question 2
>
> Next, regarding the question
>
> > And what is the statistical intuition behind it?
>
> and a related weakness comment:
>
> > ... the reason for using Gaussian distribution by giving the principle of maximum entropy, but the reason for using Truncated Gaussian seems missing and unconvincing.
>
> We emphasize that, as explained in Section 4 of our paper, $\mathcal{N}$ and $\mathcal{TN}$ are both maximum entropy (max-ent) distributions, but for different sets of initial constraints/assumptions (Dowson & Wragg, 1973). They both assume a finite mean and variance, but $\mathcal{N}$ assumes its support is $\mathbb{R}$ whereas $\mathcal{TN}$ assumes it is an interval $(l, u)$.
>
> We know that the value of the optimal heuristic $h^{\ast}$ is always larger or equal to that of an admissible heuristic $h^{\downarrow}$. Therefore, according to the max-ent principle, $h^{\ast}$ should **not** be modelled as a $\mathcal{N}$, but as a $\mathcal{TN}$ with lower bound $l=h^{\downarrow}$.
> This modelling choice results in significantly better heuristics (see Tables 1 and 2).
>
>
> ### Re: Weaknesses: More parameters?
>
> > NLL adds the prediction of $\sigma$ ... the model will predict better with more parameters.
>
> Our improvement is not due to an increase in the number of parameters.
> In order to predict $\sigma$, the neural network (NLM) needs to have two outputs, but this only affects the size of the last linear layer of the network, which is negligible compared to the size and parameters of the entire network.
> Moreover, in our preliminary experiments, doubling the total number of model parameters (NLM width) did not significantly affect the performance.
>
> The reason why our NLL loss outperforms the standard MSE is due to the underlying modelling assumptions:
> standard MSE models $h^*$ as $\mathcal{N}(\mu,1/\sqrt{2})$, which is bad;
> the "MSE with learned sigma" models it as $\mathcal{N}(\mu,\sigma)$, which is better but still bad;
> finally, our NLL models it as $\mathcal{TN}(\mu,\sigma,lb=h^{\downarrow},\infty)$, which is the best model both empirically and theoretically (max-ent distribution).
> This is easy to check by looking at Table 1 in our main paper. It can be observed that $\mathcal{TN}$ for the _fixed/none_ and _fixed/_ $h^{FF}$ configurations obtains better accuracy (_MSE_ rows) than $\mathcal{N}$ for _learn/none_ and _learn/_ $h^{FF}$.
> This means that an ML model that only predicts $\mu$ (_fixed_ configurations) and is trained with our custom NLL outperforms another model that predicts both $\mu$ and $\sigma$ (_learn_ configurations) but is trained with the standard MSE.
>
> We believe our answers address all your questions and concerns.

---

### Author Response · Authors · 2023-11-19

We want to thank the reviewers for their insightful feedback and questions.
We are encouraged by reviewers finding our paper well-motivated (_63nm, hjVN, j4wF_), enjoyable to read (_63nm, 795y, hjVN, j4wF_) and believing our proposed method outperforms the standard, gaussian-based approach employed in the literature (_63nm, LPcd, 795y, hjVN_).
Reviewer LPcd accurately summarizes our work as a _"novel approach for learning heuristics that makes principled use of admissible estimates"_.
Lastly, we highly appreciate the encouraging comments from reviewer 795y: _"This is a great paper that successfully challenges the accepted wisdom of using MSE when learning heuristics"_.

We have significantly updated our paper not only to answer reviewers' concerns, but also to provide a new enhancement that improved the performance of the proposed approach. Therefore, we kindly ask for a reassessment of our paper in light of this new information.

Below, we address some comments shared among reviewers.

Reviewers LPcd and j4wF expressed some concerns regarding the **planning results** shown in Table 2.
As reviewers 63nm, LPcd, and j4wF explicitly agreed in their comments, **our $\mathcal{TN}$ approach significantly outperforms the other methods in terms of accuracy**.
Additionally, as Table 2 shows, **$\mathcal{TN}$ also outperforms $\mathcal{N}+clip$ (and $\mathcal{N}$) in terms of planning performance** in 5 out of 6 cases, when both coverage and number of node evaluations are considered (this is needed because coverage sometimes saturates to 100%, as explained in our paper).
Nonetheless, although our best $\mathcal{TN}$ model significantly outperforms the $h^{FF}$ baseline in blocksworld, it obtains similar planning results in logistics and is outperformed by $h^{FF}$ in satellite.
This is counter-intuitive because our approach obtains significantly better MSE than $h^{FF}$ for all three domains.

At the end of section 5, we hypothesized that this happens because GBFS is sensitive to changes in node ordering caused by minor float fluctuations in heuristic values. $h^{FF}$ has no such fluctuations as it always returns an integer. It is also a strong baseline for logistics and satellite, as it obtains an MSE lower than 1 in both domains. Therefore, even though heuristics learned by $\mathcal{TN}$ are significantly more accurate _relative to_ $h^{FF}$, the _absolute_ heuristic error difference is small, leading to only a minor improvement, which is then masked by degradations from float fluctuations.

After our initial submission, we found a simple yet effective mitigation to this issue that modifies the **_tiebreaking_** strategy of GBFS, which is known to significantly affect its performance (Asai and Fukunaga, 2017).
According to our hypothesis, when the learned heuristic predicts very similar $h$ values for different nodes, the order between them (which is quite sensitive to minor changes in float values) may not be ideal because the $\mathcal{TN}$ model does not explicitly penalize wrong ordering, unlike learning-to-rank approaches. A simple mitigation is to round $h$ to an integer and then break ties with the $h^{FF}$ heuristic so that the node expansion order among similar $h$-valued nodes becomes in line with that of the $h^{FF}$ heuristic.
We have updated our paper with new tiebreaking experiments (see Appendix G).
As shown in Table S10, FF-based tiebreaking greatly improves planning results, allowing **the best $\mathcal{TN}$ configuration to also outperform the FF baseline in logistics and satellite**, when both coverage and node evaluations are considered.

Masataro Asai and Alex Fukunaga. "Exploration among and within plateaus in greedy best-first search." ICAPS. Vol. 27. 2017.

Another request (by LPcd and j4wF) was to add new experiments for evaluating the performance of our method on larger problems. **We have updated our paper with experiments that test generalization to larger problems and the results are positive**
(see Appendix H).
Tables S11 and S12 show how the NLM and LR models successfully generalize to larger problems, with only a minor decrease in accuracy for the NLM and a moderate decrease
for LR, when compared to the results for small problems.
On the other hand, HGN obtains much worse results for larger problem sizes, i.e., fails to generalize.
Since it fails even for the $\mathcal{N}$ model (which includes traditional MSE-based training), we ascribe this generalization failure to HGN. This is natural since generalization mainly depends on the neural architecture employed for learning.
Therefore, these results support our previous claims: **$\mathcal{TN}$ outperforms both $\mathcal{N}(+clip)$ and the $h^{FF}$ baselines, regardless of problem size.**

---

### Comment · Area_Chair_HKr5 · 2023-11-22

Dear reviewers,

This a reminder that deadline of author/reviewer discussion is AOE Nov 22nd (today). Please engage in the discussion, check if your concerns are addressed, and make potential adjustments to the rating and reviews.

Thank you!
AC

---

### Meta-Review · Area_Chair_HKr5 · 2023-12-05

**Metareview:**

This paper studies the problem of learning heuristic functions for forward search algorithms in planning, and points out that minimizing MSE is not the right solution. Instead, the proposed approach uses truncated Gaussians to model the learned heuristic, and uses admissible heuristics as lower bounds of the truncated Gaussians. The reviewers found the paper well-written and enjoyable to read.

Several reviews suggested that the experiments support the claims. But during discussion, reviewers reached agreement that the experiments could be more comprehensive. Moreover, 3 out of 5 reviewers expressed major concerns on novelty: "it falls short of the innovation required for acceptance solely through refining the distributional assumption for modeling"; and practicality: "After reading the appendix, I am not certain the three environments are really practical. They might be classical academic benchmarks but seem disconnected from real applications.". Reviewer j4wF also pointed out that "The tie-breaking with $h^{FF}$ variant mentioned in the rebuttal seems to be more robust in performances so I feel the authors should revise it to make that the focus. The method presented in the original paper does not consistently outperform $h^{FF}$."

Hence, I recommend to reject this paper. However, I encourage the authors to incorporate the reviewers' feedback in future revisions to make this work stronger. Potential improvements include 1) adding other planning algorithm baselines, to demonstrate the benefit of learning cost-to-go, 2) improving the clarity on novelty, 3) adding real-world planning experiments, e.g. robotics, to demonstrate practicality.

**Justification For Why Not Higher Score:**

Several reviews suggested that the experiments support the claims. But during discussion, reviewers reached agreement that the experiments could be more comprehensive. Moreover, 3 out of 5 reviewers expressed major concerns on novelty: "it falls short of the innovation required for acceptance solely through refining the distributional assumption for modeling"; and practicality: "After reading the appendix, I am not certain the three environments are really practical. They might be classical academic benchmarks but seem disconnected from real applications.". Reviewer j4wF also pointed out that "The tie-breaking with $h^{FF}$ variant mentioned in the rebuttal seems to be more robust in performances so I feel the authors should revise it to make that the focus. The method presented in the original paper does not consistently outperform $h^{FF}$."

**Justification For Why Not Lower Score:**

n/a

---

### Decision · Program_Chairs · 2024-01-16

Reject